# Leucine repeat adaptor protein 1 interacts with Dishevelled to regulate gastrulation cell movements in zebrafish

Xiao-Ning Cheng[1], Ming Shao [1], Ji-Tong Li[1], Yan-Fei Wang [1], Jing Qi[1], Zhi-Gang Xu[1] & De-Li Shi [1,2]

Gastrulation is a fundamental morphogenetic event that requires polarised cell behaviours for coordinated asymmetric cell movements. Wnt/PCP signalling plays a critical role in this process. Dishevelled is an important conserved scaffold protein that relays Wnt/PCP signals from membrane receptors to the modulation of cytoskeleton organisation. However, it remains unclear how its activity is regulated for the activation of downstream effectors. Here, we report that Lurap1 is a Dishevelled-interacting protein that regulates Wnt/PCP signalling in convergence and extension movements during vertebrate gastrulation. Its loss-of-function leads to enhanced Dishevelled membrane localisation and increased JNK activity. In maternal-zygotic *lurap1* mutant zebrafish embryos, cell polarity and directional movement are disrupted. Time-lapse analyses indicate that Lurap1, Dishevelled, and JNK functionally interact to orchestrate polarised cellular protrusive activity, and Lurap1 is required for coordinated centriole/MTOC positioning in movement cells. These findings demonstrate that Lurap1 functions to regulate cellular polarisation and motile behaviours during gastrulation movements.

[1] School of Life Sciences, Shandong University, 27, Shanda Nan Road, Jinan 250100, China. [2] Sorbonne Universités, UPMC Univ Paris 06, CNRS UMR7622, IBPS-Developmental Biology Laboratory, 75005 Paris, France. Xiao-Ning Cheng, Ming Shao and Ji-Tong Li contributed equally to this work. Correspondence and requests for materials should be addressed to Z.-G.X. (email: xuzg@sdu.edu.cn) or to D.-L.S. (email: de-li.shi@upmc.fr)

During vertebrate gastrulation, cells in different regions of the embryo undergo different types of morphogenetic movements. These fundamental developmental processes play a critical role in the formation of the three germ layers: ectoderm, mesoderm, and endoderm. In *Xenopus* and zebrafish, they mainly include epiboly, convergence and extension (CE), and directed cell migration[1–5]. In zebrafish, epiboly is the earliest morphogenetic movement that is initiated when the large yolk cell elevates into the blastoderm cells, which subsequently spread towards the vegetal pole to completely cover the yolk cell at the end of gastrulation[6,7]. CE movements occur throughout gastrulation. During these processes, lateral cells converge dorsally to narrow the germ layers, while dorsal midline cells extend along the anteroposterior axis to lengthen the embryo[1–5]. These morphogenetic movements are evolutionarily conserved and play a major role in shaping the vertebrate embryo.

The cellular and molecular mechanisms implicated in CE movements have been extensively studied, and are presently better defined. Cell intercalation that results from polarised cell behaviours produces the driving force for CE movements[1–5,8,9]. The non-canonical Wnt or planar cell polarity (Wnt/PCP) pathway plays a central role in orchestrating cellular orientations and asymmetric cell behaviours both in invertebrates and in vertebrates[9–17]. Dysfunction of Wnt/PCP signalling leads to cell movement defects during development[18–22], and has been implicated in human pathologies[23,24]. It is now well established that Wnt/PCP signalling, triggered by the interaction between Wnt ligands and Frizzled receptors, functions to modulate actin

polymerisation and cytoskeletal dynamics. The signal is relayed by Dishevelled (Dvl), which activates ROCK or Jun N-terminal kinase (JNK), depending on its association with the interaction partners[25–31]. Thus, Dvl occupies a key position in the Wnt/PCP pathway to regulate the activation of downstream effectors during asymmetric cell movements. It contains three highly conserved functional domains known as DIX, PDZ, and DEP, which are implicated in specific interaction with different partners, leading to distinct signalling outcomes[32–34]. Functional studies indicate that the PDZ and DEP domains are essential for the activation of Wnt/PCP signalling to establish and maintain cellular polarisation during gastrulation[18,35,36]. In addition, the subcellular localisation of Dvl, especially its membrane recruitment, is important for Wnt/PCP signalling in CE movements[35,37]. Therefore, the modality of Dvl interaction with its associated proteins plays a critical role in modulating its signalling function[38,39]. Nevertheless, although a substantial number of Dvl-interacting proteins have been identified[33], it remains largely unclear how the activity of Dvl in Wnt/PCP signalling is regulated during morphogenetic movements.

Lurap1 (leucine repeat adaptor protein 1), also known as Lrap35a, is an adaptor protein with two leucine-rich repeats at its N-terminal region and a PDZ-binding motif at the extreme C-terminus[40]. In cultured cells, it has been shown that Lurap1 regulates actomyosin retrograde flow and cell migration by forming a tripartite complex with myotonic dystrophy kinase-related Rac/Cdc42-binding kinase (MRCK), and the unconventional MYO18A through the leucine-rich repeats and the

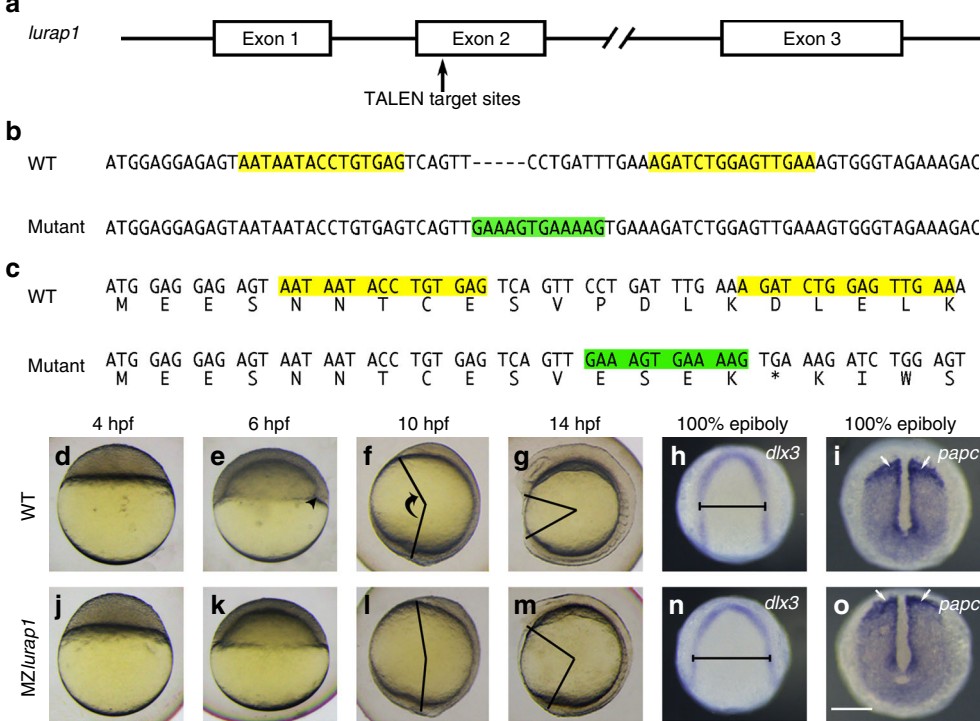

**Fig. 1** Mutation of *lurap1* in zebrafish impairs CE movements. **a** Genomic organisation of *lurap1* locus and TALEN-targeted regions. **b** Generation of an indel mutation in the second exon of *lurap1* gene following a deletion of 7 nucleotides and an insertion of 12 nucleotides (green) between TALEN recognition sequences (yellow). Dashes are introduced in the WT sequence to optimise sequence alignment. **c** Frameshift and premature termination of translation in the *lurap1* mutant transcript. **d–g** Live images of WT embryos at indicated stages, lateral view with animal pole or anterior region on the top. Arrowhead indicates the embryonic shield. The angle between the anterior end and posterior end, which reflects the extent of anteroposterior axis extension, is shown at 10 and 14 hpf. **h, i** Analysis by in situ hybridisation of *dlx3* and *papc* expression patterns shows the extent of CE movements in WT embryos at 100% epiboly stage. **j–m** Phenotypes of MZ*lurap1* embryos at indicated stages show delayed embryonic shield formation and reduced anteroposterior axis elongation. **n, o** The expression patterns of *dlx3* and *papc* in MZ*lurap1* embryos indicate defective CE movements. The neural plate is wider (horizontal line) and paraxial mesoderm is compressed (arrows). Scale bar: **d–o** 200 μm

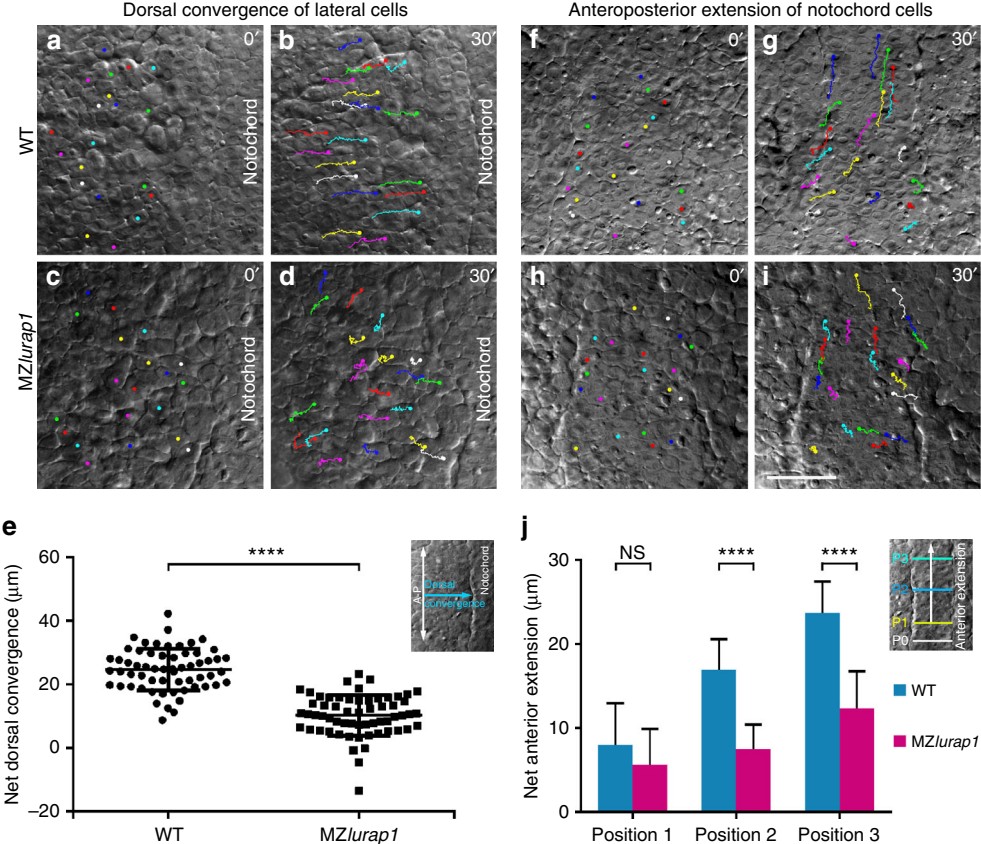

**Fig. 2** Live time-lapse analysis of cell movements in WT and MZ*lurap1* embryos. The initial and final positions of 20 randomly selected cells are labelled by coloured dots, and their movement trajectories are traced. Representative images selected from time-lapse movies are positioned with the anterior region on the top. **a**, **b** First and last images show the convergence of lateral cells towards the notochord in a WT embryo. **c**, **d** First and last images show the movement behaviours of lateral cells in a MZ*lurap1* embryo. **e** Scatter plot shows the net distance, represented by the horizontal arrow in the inset, reached by lateral cells towards the notochord during convergence in WT and MZ*lurap1* embryos. Bars represent the mean values ± s.d. from three independent experiments (****$P < 0.0001$; Student's $t$-test). **f**, **g** First and last images show the anterior extension of notochord cells in a WT embryo. **h**, **i** First and last images show the movement behaviours of notochord cells in a MZ*lurap1* embryo. **j** Graph shows the net anterior extension, represented by the vertical arrow in the inset, reached by notochord cells located at three different anteroposterior positions, as represented by the coloured horizontal lines in the inset. Bars represent the mean values ± s.d. from three independent experiments (****$P < 0.0001$; NS, not significant; Student's $t$-test). The three positions (P1, P2, and P3) are selected at the middle region of the anteroposterior length in both WT and MZ*lurap1* embryos, and located at a distance of 20, 50, and 80 μm anterior to the arbitrary reference position (P0), respectively. Scale bar: **a–d**, **f–i** 50 μm

PDZ-binding motif, respectively[40,41]. Although this protein is highly conserved among vertebrate species, its implication in regulating cell movements during early development has never been reported.

Here we show that Lurap1 is required for CE movements in vertebrate embryos. Both loss-of-function and gain-of-function of Lurap1 produce characteristic defective CE movements. We find that Lurap1 physically interacts with the PDZ domain of Dvl through its PDZ-binding motif, and limits Dvl membrane localisation and JNK activation. In zebrafish maternal-zygotic *lurap1* (MZ*lurap1*) mutants, polarised cellular protrusions and directional cell movements are defective in both dorsal midline and lateral regions of the embryo during gastrulation. Coordinated positioning of the centriole/MTOC (microtubule organisation centre), a conserved readout of Wnt/PCP signalling, is also disrupted in axial mesoderm and neuroectoderm. Moreover, we show that Lurap1, Dvl, and JNK functionally interact to establish cellular polarisation. Collectively, our results identify Lurap1 as an interaction partner of Dvl, and suggest that it functions in the Wnt/PCP pathway to regulate cytoskeleton organisation and cell polarity. This study provides further insight into the molecular mechanism that orchestrates

motile cell behaviours during CE movements in vertebrate embryos.

## Results

**Mutation of *lurap1* affects CE movements**. Zebrafish genome contains a single copy of *lurap1* gene according to the Zv9 assembly (www.ensembl.org). It is located on chromosome 6 and comprises three exons (Fig. 1a). At the protein level, both the leucine-rich repeats and the C-terminus are highly conserved in human and zebrafish Lurap1 (Supplementary Fig. 1A). In situ hybridisation analysis indicated that maternal *lurap1* transcripts are present both in the blastomeres and the yolk cell, and zygotic *lurap1* is expressed ubiquitously during gastrulation and subsequent stages (Supplementary Fig. 1B–E). This implies that *lurap1* may play an early role during development. To analyse its function, we used transcription activator-like effector nucleases (TALENs) genome-editing approach to generate *lurap1* mutant line, and obtained a five base pair insertion in the second exon (Fig. 1b). This results in a frameshift after the valine residue at position 11 and premature termination of translation (Fig. 1c). RT-PCR followed by sequence analysis confirmed the mutation at

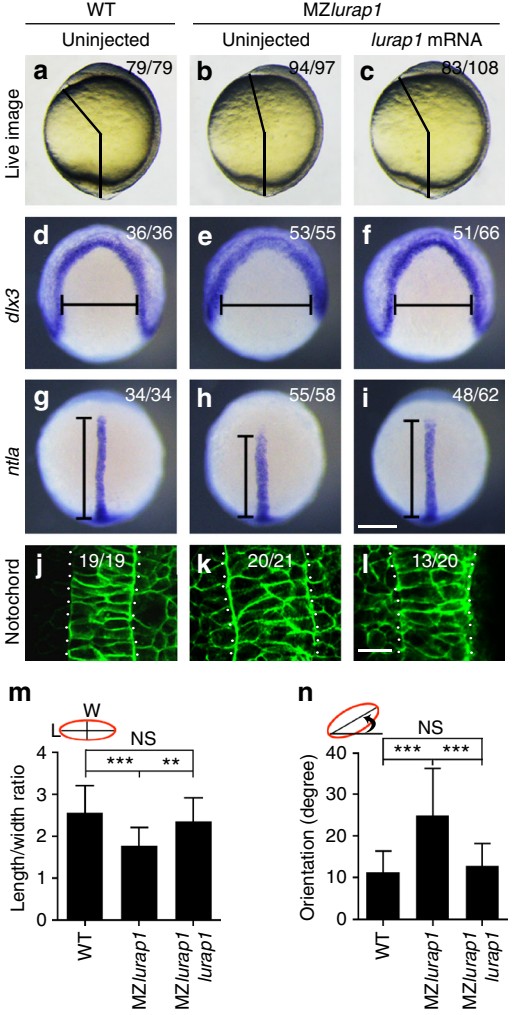

**Fig. 3** Lurap1 rescues MZ*lurap1* mutant phenotypes and notochord cell polarity. **a**–**c** Phenotype rescue is determined by measuring the angle between the anterior end and posterior end in indicated embryos. **d**–**i** Analysis by in situ hybridisation of *dlx3* and *ntla* expression patterns shows the rescue of neural plate convergence (horizontal line) and notochord extension (vertical line) in MZ*lurap1* embryos. **j**–**l** Analysis by confocal microscopy shows the rescue of notochord width and cell shape in MZ*lurap1* embryos. **m**, **n** Graphs show rescue of the LWR and the orientation of notochord cells in MZ*lurap1* embryos. The data were calculated using a total of 135 cells from 9 embryos in each condition. Bars represent the mean values ± s.d. from three independent experiments (**$P < 0.01$; ***$P < 0.001$; NS, not significant; Student's *t*-test). Scale bars: **a**–**i** 200 μm; **j**–**l** 20 μm

the mRNA level (Supplementary Fig. 2), suggesting a positive targeting effect that generates a loss-of-function of *lurap1* gene.

Heterozygous mutant embryos developed normally, and zygotic homozygous mutant embryos exhibited weak CE defects. MZ*lurap1* embryos, however, displayed obvious developmental defects that were morphologically discernible at 4 h post fertilisation (hpf). At this stage, the border between the blastodisc and the yolk cell in MZ*lurap1* embryos was not as flat as in wild-type (WT) embryos (Fig. 1d, j). At 6 hpf (50% epiboly) when the embryonic shield was clearly visible in WT embryos (Fig. 1e), no thickening of the germ ring could be observed in MZ*lurap1* embryos (Fig. 1k), indicating impaired convergence of lateral cells towards the dorsal region. However, dorsoventral patterning was unaffected (Supplementary Fig. 3). During gastrulation,

MZ*lurap1* embryos showed defective CE phenotypes. At 10 hpf when WT embryos completed epiboly, MZ*lurap1* embryos exhibited axis extension defect with reduced anteroposterior elongation, as reflected by the angle formed between the anterior end and posterior end of the embryo (Fig. 1f, l). At 14 hpf, the defective axis extension was more prominent. While WT embryos formed a long anteroposterior axis with brain rudiment and tail bud clearly visible, MZ*lurap1* embryos displayed an obviously reduced anteroposterior length (Fig. 1g, m). The CE defects were further confirmed by in situ hybridisation analysis when both WT and MZ*lurap1* embryos reached 100% epiboly. Using *dlx3* and *papc* as markers, which label neural plate borders and paraxial mesoderm, respectively, we found that, compared with WT embryos (Fig. 1h, i), MZ*lurap1* embryos exhibited broader neural plate and compressed paraxial mesoderm (Fig. 1n, o), indicating CE defects. Moreover, simultaneous in situ hybridisation analysis of *pax2a*, *krox20*, and *myod1* expression patterns in flat-mounted embryos revealed that paraxial mesoderm in MZ*lurap1* embryos was wider compared to WT embryos with an equal number of somites (Supplementary Fig. 4). This indicates that the CE defects were not caused by a developmental delay, and that maternal Lurap1 is specifically required for CE movements.

We then analysed cell movement behaviours by time-lapse imaging and individual cell tracing. At 90% epiboly, the dorsal convergence of lateral cells and anteroposterior extension of notochord cells were recorded using differential interference contrast microscopy for a period of 30 min. In WT embryos, lateral cells moved towards the notochord with regular trajectories, reaching an average net distance of $24.75 \pm 0.85$ μm ($n = 60$ from three embryos). By contrast, in MZ*lurap1* embryos, these cells moved along irregular or zigzagging trajectories, and reached an average net distance of $10.29 \pm 0.83$ μm ($n = 60$ from three embryos), suggesting impaired convergence (Fig. 2a–e; Supplementary Movies 1 and 2). Compared with WT embryos, notochord cells in MZ*lurap1* embryos also displayed irregular movement trajectories and reduced anterior extension (Fig. 2f–i; Supplementary Movies 3 and 4). When notochord cells at three different anteroposterior levels were selected to measure their net movement distance ($n = 30$ cells from three embryos in each condition), we found that MZ*lurap1* cells were less effective to move anteriorly than WT cells at the same position (Fig. 2j). For example, WT cells located at 80 μm (P3) anterior to the reference position (P0) moved an average net distance of $23.72 \pm 3.69$ μm, whereas MZ*lurap1* cells only reached $12.30 \pm 4.49$ μm, indicating impaired anterior extension. These results clearly show that mutation of *lurap1* affects CE movements during gastrulation.

To ascertain that the CE defects were specifically caused by loss-of-function of *lurap1*, we performed rescue experiments by injecting *lurap1* mRNA (100 pg) into MZ*lurap1* embryos at 1-cell stage. Compared with WT embryos (Fig. 3a), nearly all uninjected MZ*lurap1* embryos (94/97) showed reduced anteroposterior elongation at 10 hpf (Fig. 3b), whereas a large majority of *lurap1*-injected MZ*lurap1* embryos (83/108) were indistinguishable from WT embryos (Fig. 3c). A similar extent of rescue of neural plate width (51/66) and notochord length (48/62) was also observed by in situ hybridisation analysis of *dlx3* and *ntla* expression patterns (Fig. 3d–i). We then labelled the embryos with membrane GFP (mGFP), and analysed notochord width and cell shape by confocal microscopy (Fig. 3j–l). The notochord displayed two cell layers in WT embryos at 90–100% epiboly (19/19), whereas it was wider, with approximately three cell layers in nearly all examined MZ*lurap1* embryos (20/21) at the same stage. Injection of *lurap1* mRNA restored notochord width in a majority of MZ*lurap1* embryos (13/20). It also rescued the length/width ratio (LWR) and the mediolateral orientation of MZ*lurap1* notochord cells (Fig. 3m, n). Furthermore, time-lapse analysis of

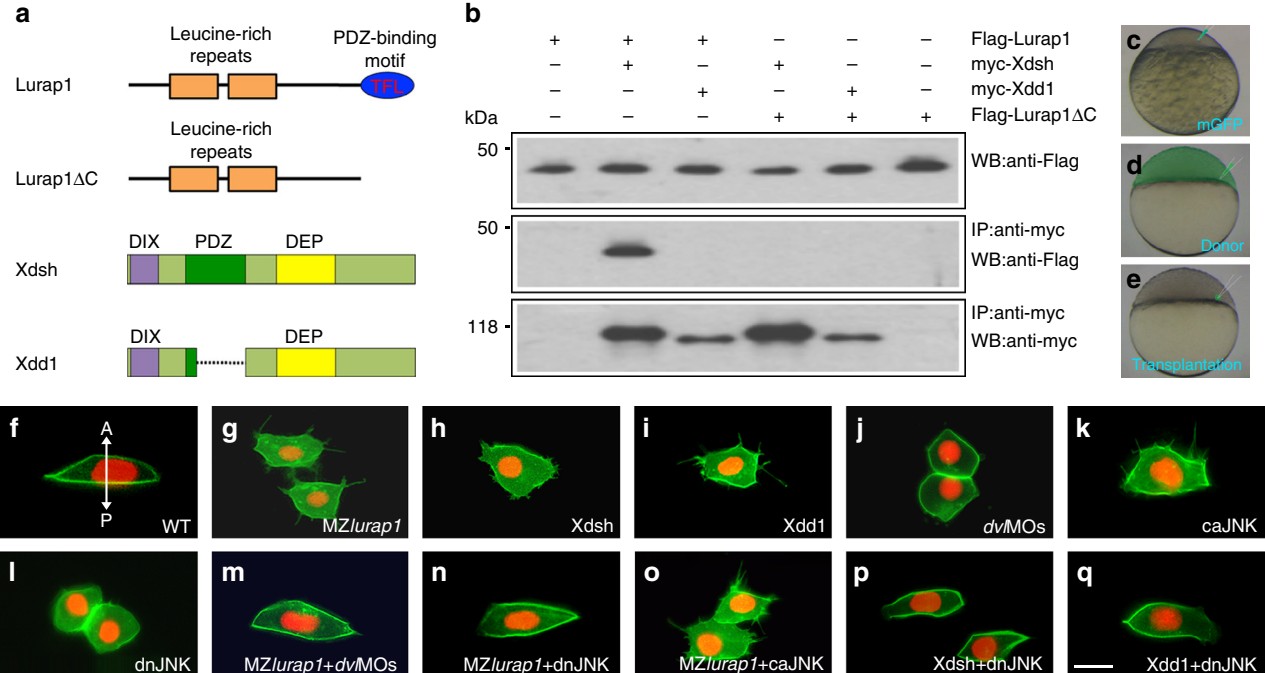

**Fig. 4** Lurap1 interacts with Dvl and JNK in cellular polarisation. **a** Schematic representation of Lurap1 and Dvl proteins. **b** Co-immunoprecipitation shows physical interaction between Lurap1 and Dvl. **c–e** Cell transplantation procedure for the analysis of cell polarity and protrusive activity. Labelled donors and unlabelled recipient embryos have either the same genotype, or they are injected with identical mRNAs or morpholinos. **f–q** Polarised behaviours and protrusive activities of transplanted cells integrated in the notochord of recipient embryos. The cell membrane is labelled by mGFP and the nucleus is labelled by Histone2B-RFP. The experiments were performed using at least three independent offspring, and a total of at least 10 embryos were analysed in each condition. Representative images are shown, with the anterior region positioned on the top. **f** A bipolar and mediolaterally elongated WT cell with unidirectional filopodia. The bidirectional arrow indicates anterior (A) and posterior (P) axis. **g** Multipolar protrusive MZ*lurap1* mutant cells are less elongated, with lamellipodia and filopodia in all directions. **h** A Xdsh-overexpressing cell loses polarity and forms multiple lamellipodia that also extend filopodia in random directions. **i** A Xdd1-overexpressing cell becomes multipolar protrusive, with extensive filopodia in all directions. **j** Knockdown of *dvl2* and *dvl3a* in WT cells blocks protrusive activity, and morphant cells take a round shape. **k** Overexpression of caJNK in a WT cell disrupts polarised cell behaviours and induces multiple filopodia in random directions. **l** Blockade of JNK signalling in WT cells results in loss of protrusive activity and cell polarity. **m–o** The mediolateral polarity and protrusive activity of MZ*lurap1* mutant cells are rescued by reducing Dvl dosage or JNK activity, but not by increasing JNK activity. **p**, **q** Reducing JNK activity in Xdsh-overexpressing or Xdd1-overexpressing cells restores mediolateral polarity, and prevents the formation of randomised lamellipodia and filopodia. Scale bar: **f–q** 20 μm

cell movement behaviours indicated that Lurap1 efficiently rescued CE movements in MZ*lurap1* embryos (Supplementary Fig. 5; Supplementary Movies 5 and 6). Thus, we conclude that loss-of-function of *lurap1* in zebrafish specifically disrupts directional cell movements and polarised cell behaviours.

**Lurap1 interacts with Dvl and JNK in cellular polarisation.** Since MZ*lurap1* embryos exhibited characteristic CE defective phenotypes reminiscent of perturbed Wnt/PCP signalling, and a conserved class I PDZ-binding motif (T-F-L) is present at the extreme C-terminus of Lurap1, we wondered whether there exists a possible interaction between Lurap1 and Dvl. By co-immunoprecipitation assay, we found that the full-length Lurap1, but not the truncated Lurap1ΔC lacking the PDZ-binding motif, co-immunoprecipitated with the full-length *Xenopus* Dvl2 (Xdsh) following transfection in HEK293 cells. However, the full-length Lurap1 did not co-immunoprecipitate with Xdd1, which lacks the PDZ domain (Fig. 4a, b; Supplementary Fig. 6). This clearly indicates a physical interaction between Lurap1 and Dvl, which requires both the C-terminus PDZ-binding motif of Lurap1 and the PDZ domain of Dvl.

The next issue is to examine whether Lurap1 and Dvl functionally interact in CE movements. We first analysed the extent of CE defects in MZ*lurap1* embryos overexpressing Xdd1, by following the convergence of lateral cells and the extension of

dorsal cells after photo-conversion of Kaede GFP in a small group of cells. Statistical analysis (unpaired Student's *t*-test) revealed that injection of a low dose of *Xdd1* mRNA (80 pg) elicited mild defective CE phenotypes in WT embryos, but it strongly enhanced CE defects in MZ*lurap1* embryos (Supplementary Fig. 7). This indicates a functional interaction between Lurap1 and Wnt/PCP signalling in CE movements. However, the modality of Lurap1 in Wnt/PCP signalling remains to be determined. Xdd1 lacking the PDZ domain is inactive in Wnt/ß-catenin signalling[18], but the DEP domain has been shown to be sufficient for JNK activation[42,43], suggesting that the CE defects caused by Xdd1 overexpression may be due to an increased Wnt/PCP signalling[44,45]. Since both inappropriate activation and inhibition of the Wnt/PCP pathway perturb cell polarity[20], the amplification of CE defects by Xdd1 in MZ*lurap1* embryos implies that loss-of-function of *lurap1* would cause an upregulation of Wnt/PCP activity. Thus Lurap1 should normally function to limit the activation of Wnt/PCP signalling.

To further verify this possibility, we examined how Lurap1 and Dvl functionally interact to establish polarised cell behaviours in CE movements. In order to better visualise cellular protrusions, we transplanted mGFP-labelled cells to an unlabelled recipient embryo at sphere stage (Fig. 4c–e). When chimeric embryos reached 90% epiboly stage, polarity changes of labelled cells located in the notochord were analysed by time-lapse imaging.

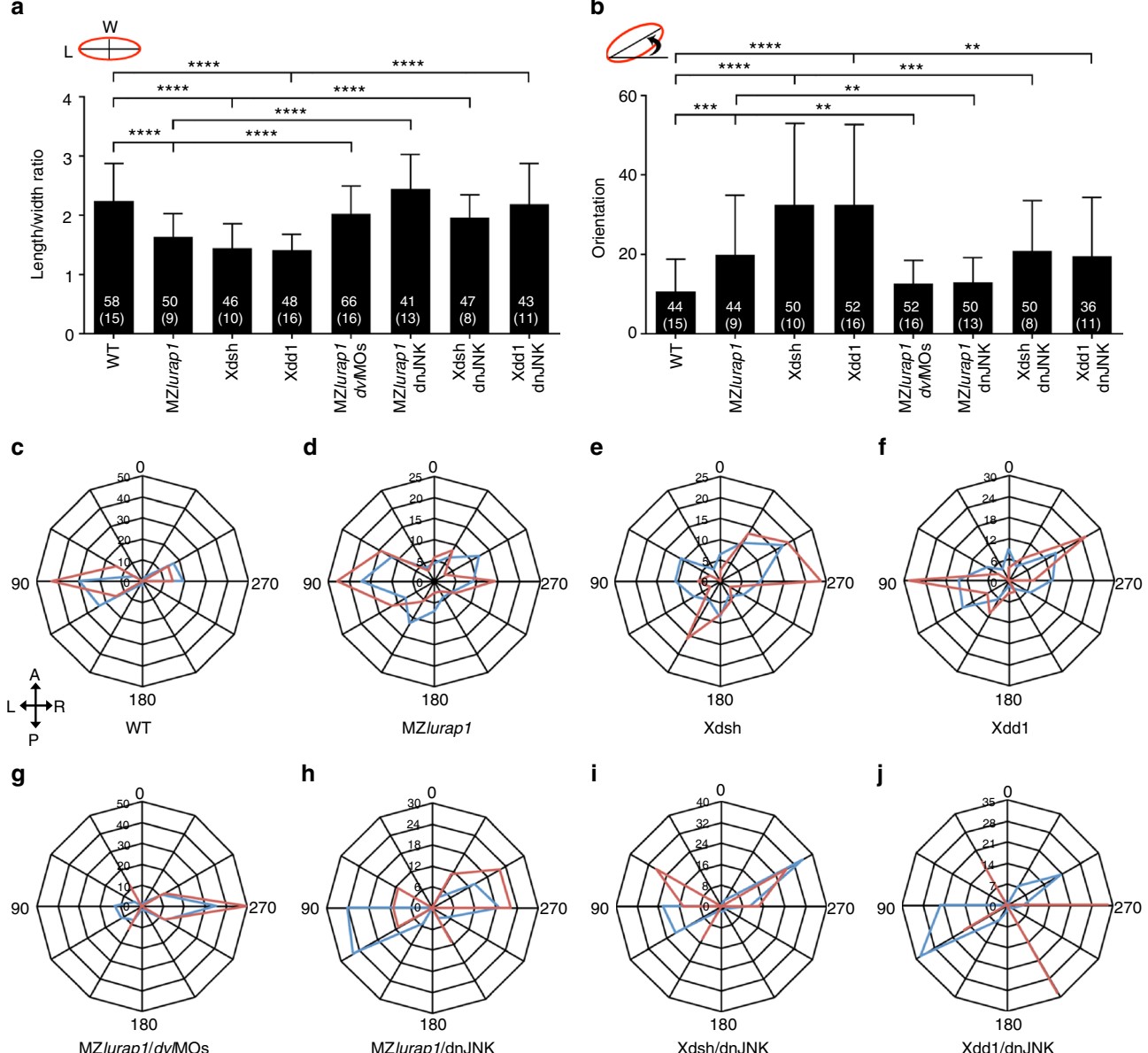

**Fig. 5** Statistical analyses of Lurap1 interaction with Dvl and JNK in cell polarity. mGFP-labelled cells were transplanted in the notochord of recipient embryos, and chimeric embryos were cultured to at 90–100% epiboly. Live images were collected, and different parameters of cell polarity were analysed using ImageJ software. **a**, **b** Analyses of the cell shape (LWR) and orientation from indicated conditions. Cell polarity defects in MZ*lurap1* mutant cells, and in Xdsh-overexpressing or Xdd1-overexpressing cells are rescued by reducing Dvl dosage or JNK activity. Bars represent the mean values ± s.d. from three independent experiments (**$P < 0.01$; ***$P < 0.001$; ****$P < 0.0001$; Student's $t$-test). Numbers in each bar represent total cells analysed, with total embryos indicated in parentheses. **c–j** Analyses of the types and orientation of lamellipodia (red) and filopodia (blue) in indicated conditions. The orientation of lamellipodia and filopodia was determined by measuring the angle between the anteroposterior axis passing through the cell nucleus, clockwise to the protruding direction. The bidirectional arrows indicate anterior (A) and posterior (P) axis, and left (L) and right (R) axis of the embryo. The defective polarised protrusive activity in MZ*lurap1* mutant cells, and in Xdsh-overexpressing or Xdd1-overexpressing cells is rescued by reducing Dvl dosage or JNK activity. Note that, in WT and *dvl*MOs-rescued MZ*lurap1* mutant cells, the predominant presence of cell protrusions on one side of the cells is due to the attachment of the other side to the notochord boundary. The data were obtained from three independent experiments using at least 30 cells derived from 8 to 10 embryos. The number on each ring of the radar graph represents the percentage of cell protrusions

We found that WT cells were bipolar and elongated, with cellular protrusions, essentially filopodia, only in the mediolateral direction (Fig. 4f; Supplementary Movie 7). By contrast, MZ*lurap1* mutant cells in the notochord were not correctly aligned mediolaterally, and were less elongated. They displayed unstable multipolar protrusive activity by forming and retracting lamellipodia, and especially very long filopodia in all directions (Fig. 4g; Supplementary Movie 8). Interestingly, MZ*lurap1* cells located in the ectoderm were round with multiple blebs in randomised directions (Supplementary Movie 9), indicating increased, but undirected motility. Similarly, overexpression of Xdsh and Xdd1 by injecting the corresponding mRNA (80 pg) in WT embryos produced multipolar protrusive cells. Xdsh induced more lamellipodia in all directions, which also extended filopodia, whereas Xdd1 essentially induced randomised filopodia (Fig. 4h, i; Supplementary Movies 10 and 11). This difference may be due to their activity to interact with downstream effectors. Conversely, knockdown of the most abundantly expressed Dvl2 and Dvl3a

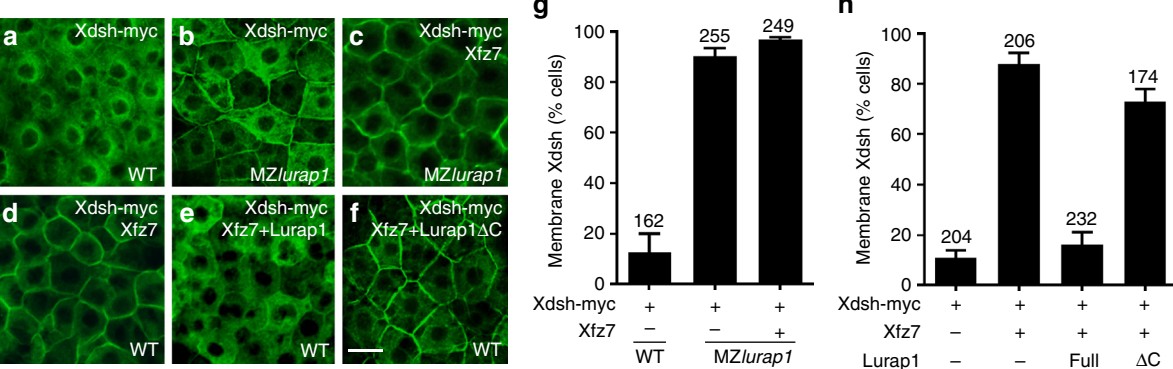

**Fig. 6** Lurap1 modulates Dvl membrane localisation. Analysis by confocal microscopy of myc-tagged Xdsh subcellular localisation in indicated conditions. **a** Xdsh is localised predominantly in the cytoplasm, and weakly at the plasma membrane in a WT embryo. **b** Enhanced Xdsh membrane localisation in the blastoderm cells of a MZ*lurap1* embryo. **c** Xfz7 further enhances Xdsh membrane localisation in a MZ*lurap1* embryo. **d** Xdsh membrane recruitment by Xfz7 in WT embryo. **e** Lurap1 prevents Xdsh membrane recruitment by Xfz7. **f** Lurap1ΔC has no effect on Xdsh membrane recruitment by Xfz7. **g**, **h** Graphs show the extent of Xdsh membrane localisation in different conditions. The cells with stronger or equal membrane than cytoplasmic Xdsh staining are counted. Numbers on the top of each bar indicate total cells examined from 6 to 8 embryos in three independent experiments. Scale bar: **a**–**f** 20 μm

isoforms[46], using high dose of the corresponding morpholinos (4 and 10 ng per embryo, respectively), produced CE defects (Supplementary Fig. 8), and disrupted cellular polarisation by blocking the formation of lamellipodia and filopodia (Fig. 4j; Supplementary Movie 12). Consistent with the activation of JNK by Dvl in Wnt/PCP signalling[42,43], overexpression of a constitutively active form of JNK (caJNK) by injecting the corresponding mRNA (100 pg) in WT embryos disrupted cell polarity by inducing randomised filopodia, whereas blocking JNK activity by injection of mRNA (300 pg) coding for the dominant negative JNK mutant (dnJNK) produced rounded cells lacking cellular protrusions (Fig. 4k, l; Supplementary Movies 13 and 14). Furthermore, injection of low doses of *dvl2* and *dvl3a* morpholinos (1 and 2.5 ng per embryo, respectively) in MZ*lurap1* embryos rescued cell shape and mediolateral orientation, and restored the bipolar protrusive activity by reducing randomised lamellipodia and filopodia (Figs. 4m, 5; Supplementary Fig. 9; Supplementary Movie 15). Similarly, injection of a low amount of *dnJNK* mRNA (100 pg), but not *caJNK* mRNA, also rescued cell polarity in MZ*lurap1* mutants, and in Xdsh-overexpressed or Xdd1-overexpressed embryos (Figs. 4o–q, 5; Supplementary Fig. 9; Supplementary Movies 16–18). These observations are consistent with the control of cell polarity by Dvl during gastrulation[36], and indicate that Lurap1 functions upstream of JNK signalling to regulate Dvl activity in polarised protrusive activity during CE movements.

**Lurap1 regulates Dvl membrane localisation**. Membrane targeting of Dvl was shown to be required for vertebrate Wnt/PCP signalling and CE movements[37]. To determine how Lurap1 regulates Dvl function in CE movements, we analysed Dvl membrane localisation following Lurap1 loss-of-function and gain-of-function. We first examined the subcellular localisation of myc-tagged Lurap1 in neuroectoderm, paraxial mesoderm, and notochord cells at 90% epiboly by confocal microscopy. The result showed that it was present in the cytoplasm, with localised distribution in some cells of WT embryos (Supplementary Fig. 10A–C). However, immunostaining staining of myc-tagged Lurap1 seemed to be more diffuse and uniform in the cytoplasm of most cells in *dvl2/dvl3a* morphants or *vangl2* zygotic homozygous mutants (Supplementary Fig. 10D–I). We then injected myc-tagged *Xdsh* mRNA (100 pg) into WT or MZ*lurap1* embryos, with or without *Xfz7* mRNA (150 pg) coding for the *Xenopus* Frizzled7, which recruits Xdsh to the plasma membrane[47]. In WT embryos, Xdsh was mainly localised in the cytoplasm of blastoderm cells, with weak membrane localisation (Fig. 6a, g). In MZ*lurap1* embryos, however, nearly all cells showed substantial Xdsh membrane localisation (Fig. 6b, g). Coexpression of Xfz7 with Xdsh in MZ*lurap1* embryos further increased the intensity, and the number of cells with Xdsh membrane localisation (Fig. 6c, g). Conversely, overexpression of Flag-tagged Lurap1, but not Lurap1ΔC, in WT embryos by injecting the corresponding mRNA (200 pg) strongly prevented Xfz7-mediated Xdsh membrane recruitment (Fig. 6d–f, h). These indicate that Lurap1 may function to regulate the membrane localisation of Dvl through its PDZ-binding motif, and that it likely prevents Dvl membrane recruitment by retaining it in the cytoplasm. Because Dvl membrane localisation is required for Wnt/PCP signalling[35,37], the results suggest that Lurap1 may negatively regulate the activation of the Wnt/PCP pathway.

**Overexpression of Lurap1 affects CE movements**. Since Lurap1 could prevent Dvl membrane recruitment, we injected *lurap1* or *lurap1ΔC* mRNA (200 pg) in WT zebrafish embryos at 1-cell stage to see whether its overexpression also produces CE defects. In situ hybridisation analysis of *dlx3* and *ntla* expression patterns at bud stage showed that *lurap1*-injected embryos displayed wider neural plate and shortened notochord (Fig. 7a, b, d, e), indicating CE defects. However, a majority of embryos overexpressing Lurap1ΔC showed normal *dlx3* and *ntla* expression patterns (Fig. 7a, c, d, f). Moreover, analysis by confocal microscopy indicated that overexpression of Lurap1, but not Lurap1ΔC, in WT embryos increased notochord width (Fig. 7g–i), and perturbed the LWR and mediolateral polarity of notochord cells (Fig. 7j, k). Similar results were obtained in *Xenopus* by injecting different amounts of *lurap1* or *lurap1ΔC* mRNA (200 and 600 pg) in the dorsal equatorial region at 4-cell stage. They showed that overexpression of Lurap1 produced characteristic CE phenotypes since injected embryos displayed bent and shortened anteroposterior axis at tail-bud stages (Fig. 7l–n). The effect of Lurap1 overexpression on CE movements was dose-dependent, and was comparable with that of Xdd1 (Fig. 7o). Injection of a high amount of *lurap1ΔC* mRNA (600 pg) produced weak CE defects (Fig. 7o), implying that other regions in Lurap1, such as the leucine-rich repeats that bind and activate MRCK[40], may play a role in cell movements. Together, these observations further suggest that Lurap1 is implicated in regulating cell polarity during CE movements.

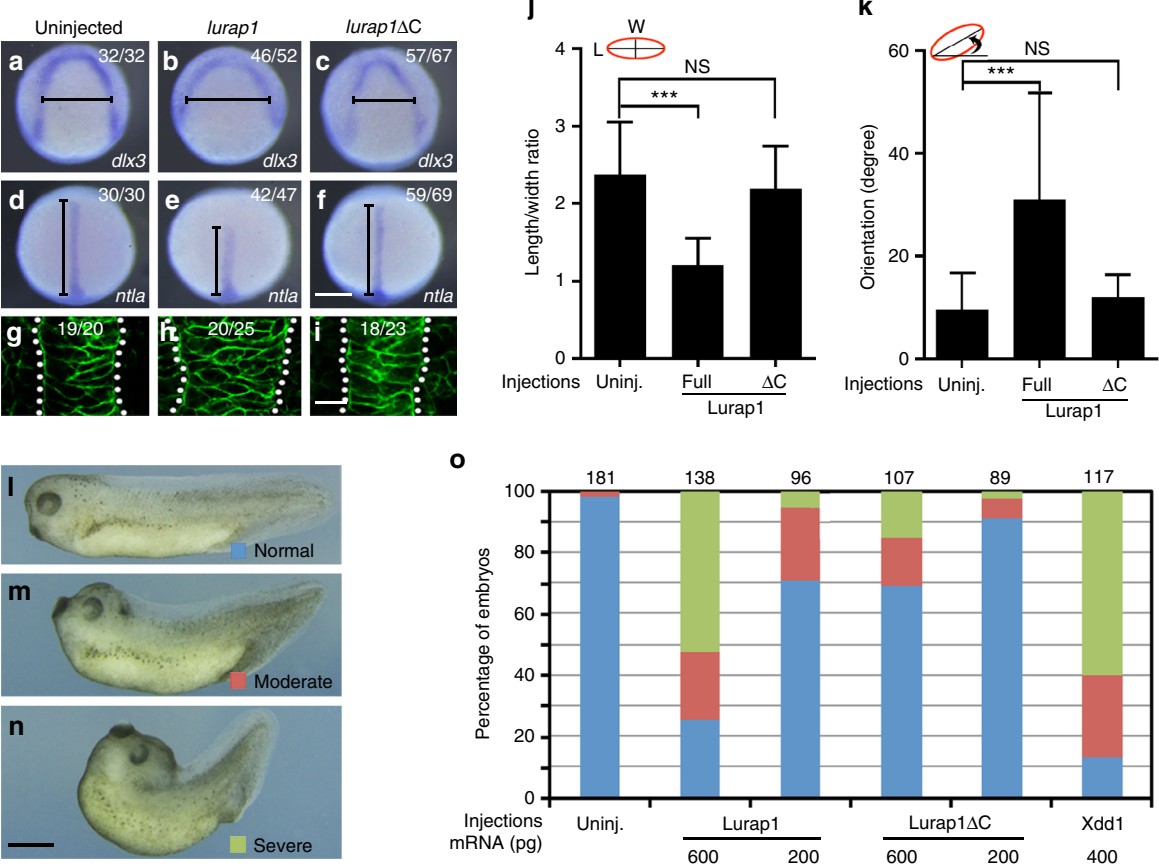

**Fig. 7** Overexpression of Lurap1 impairs CE movements and notochord cell polarity. **a–f** In situ hybridisation analysis using *dlx3* (**a–c**) and *ntla* (**d–f**) to reflect the extent of neural plate convergence and axial mesoderm extension in zebrafish embryos. Overexpression of Lurap1 potently affects CE movements, while Lurap1ΔC shows a weak effect. **g–i** Analysis by confocal microscopy of notochord width and cell shape in indicated conditions. Representative images are shown, with statistical numbers of embryos scored from three independent experiments. **j**, **k** Graphs show the statistics of cell shape (LWR) and orientation of notochord cells in uninjected, and *lurap1* or *lurap1ΔC*-injected embryos. Bars represent the mean values ± s.d. and the data were calculated using 20 cells randomly selected in a representative image for each condition (***$P < 0.001$; NS, not significant; Student's *t*-test). **l–n** Live images of normal and CE defective *Xenopus* embryos at tail-bud stage. **o** Statistical analysis of the dose-dependent CE defects following overexpression of Lurap1 or Lurap1ΔC. The colour codes for different categories of phenotypes are indicated in the live images on the left. The data were scored from four experiments using different batches of embryos, with total number of embryos shown on the top of each stacked column. Xdd1 was included in the analysis for a comparison. Scale bars: **a–f** 200 μm; **g–i** 20 μm; **l–n** 1.2 mm

**Mutation of *lurap1* disrupts MTOC positioning**. Wnt/PCP signalling has been shown to orchestrate polarised cell behaviours by controlling the position of the centriole/MTOC both in vertebrates and in invertebrates[48,49], thereby influencing microtubule cytoskeleton. We thus analysed the position of MTOCs relative to the cell nucleus, and with respect to the anteroposterior axis of the embryo at 90% epiboly. This was performed through triple staining by centrin4-GFP to localise MTOCs, DAPI staining of the nucleus, and membrane RFP (mRFP) staining to outline the cell surface. Consistent with previous observation[48], statistical analyses (unpaired Student's *t*-test) of more than 100 cells from three independent experiments using different batches of embryos revealed that MTOCs displayed a biased location in both neuroectoderm and notochord cells of WT embryos. They were mostly located at the posterior positions of the nucleus in neuroectoderm cells (Fig. 8a, d), or at the posterior and lateral positions of the nucleus in notochord cells (Fig. 8g, j). In MZ*lurap1* mutants, MTOCs were shifted to more lateral positions of the nucleus in neuroectoderm cells (Fig. 8b, e), and became randomly positioned with respect to the nucleus in notochord cells (Fig. 8h, k), indicating a disruption of asymmetric cell behaviours.

To see how inappropriate activation of JNK signalling affects MTOC positioning, we overexpressed caJNK in WT embryos by injecting the corresponding mRNA (100 pg). We found that, as in MZ*lurap1* mutants, MTOCs were essentially located at the lateral positions of the nucleus in neuroectoderm cells (Fig. 8c, f), and randomly positioned with respect to the nucleus in notochord cells (Fig. 8i, l). Furthermore, live time-lapse imaging revealed that MTOCs in notochord cells of MZ*lurap1* mutants or caJNK-overexpressing embryos were frequently changing positions with respect to the nucleus, either in anteroposterior or mediolateral direction (Fig. 8m–o; Supplementary Movies 19–21). These indicate that loss-of-function of Lurap1 or inappropriate activation of JNK signalling similarly disrupts the biased positioning of MTOCs in cells undergoing CE movements. Since the intracellular position of MTOCs organise microtubule cytoskeleton and reflects the polarity of cells engaged in asymmetric movements[48,49], these observations suggest that Lurap1 could function to establish polarised cell behaviours during CE movements by modulating cytoskeleton organisation.

**Lurap1 negatively regulates JNK activation**. To gain further insight into the mechanism by which Lurap1 regulates CE

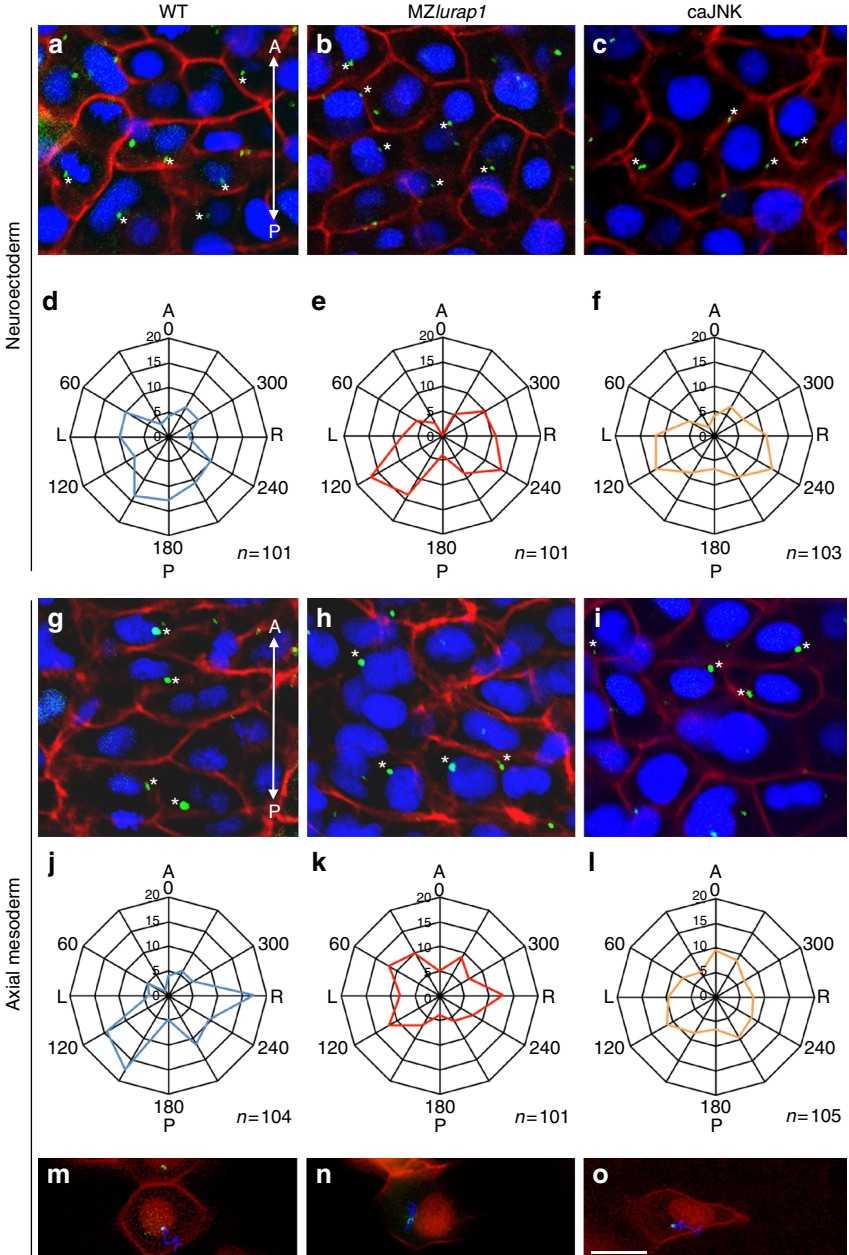

**Fig. 8** Positioning of MTOCs is affected in MZ*lurap1* mutants and caJNK-overexpressing embryos. **a–l** Immunofluorescence confocal microscopic images and statistical analysis on fixed embryos at 90% epiboly. The centriole, nucleus, and cell membrane are labelled by centrin4-GFP, DAPI, and mRFP, respectively. The position of the centriole is determined by measurement of the angle formed by the line connecting the centre of the nucleus and the centriole, relative to the anteroposterior axis passing through the nucleus. The statistical data (Student's *t*-test) are presented as radar graphs, and the number on each ring represents the percentage of centrioles. The total number of cells (*n*) examined from three different batches of embryos is indicated on the bottom right of each graph. **a–c** Location of centrioles (stars) in neuroectoderm cells of indicated embryos. The images are positioned with anterior (A) on the top, and posterior (P) towards the bottom. **d–f** Statistical analyses (Student's *t*-test) show the positions of centrioles in neuroectoderm cells. The centrioles in WT embryos display biased location at the posterior position of the nucleus. They shift to more lateral region of the nucleus in MZ*lurap1* and caJNK-overexpressing embryos, with some degrees of left (L) and right (R) symmetry. **g–i** Location of centrioles (stars) in notochord cells of indicated embryos. **j–l** Statistical analyses (Student's *t*-test) show the preferential location of centrioles at the posterior and lateral positions of the nucleus in notochord cells of WT embryos, and randomised location of centrioles in MZ*lurap1* and caJNK-overexpressing embryos. **m–o** Representative images from live time-lapse imaging shows the movement trajectories of centrioles in notochord cells of indicated embryos at 90% epiboly. For convenience, both the nucleus and the cell membrane were all labelled in red, by Histone2B-RFP and mRFP, respectively. Scale bar: **a–c**, **g–i**, **m–o** 20 μm

movements, we examined the functional interaction between Lurap1, Dvl, and JNK both in zebrafish and *Xenopus* embryos. As assayed by in situ hybridisation using *ntla* as a notochord marker, injection of a low dose of *caJNK* mRNA (50 pg) or *dnJNK* mRNA (100 pg) in WT zebrafish embryos did not produce obvious CE defects. However, caJNK enhanced, whereas dnJNK rescued CE

defects in MZ*lurap1* embryos (Supplementary Fig. 11). Since Wnt/PCP signalling functions through Dvl and JNK to activate ATF2 (activating transcription factor 2)[30], or AP1 (activator protein 1)[31], we then used the ATF2 luciferase reporter to monitor more quantitatively Lurap1-regulated JNK activity[50]. Consistent with the results from phenotype analyses, injection of

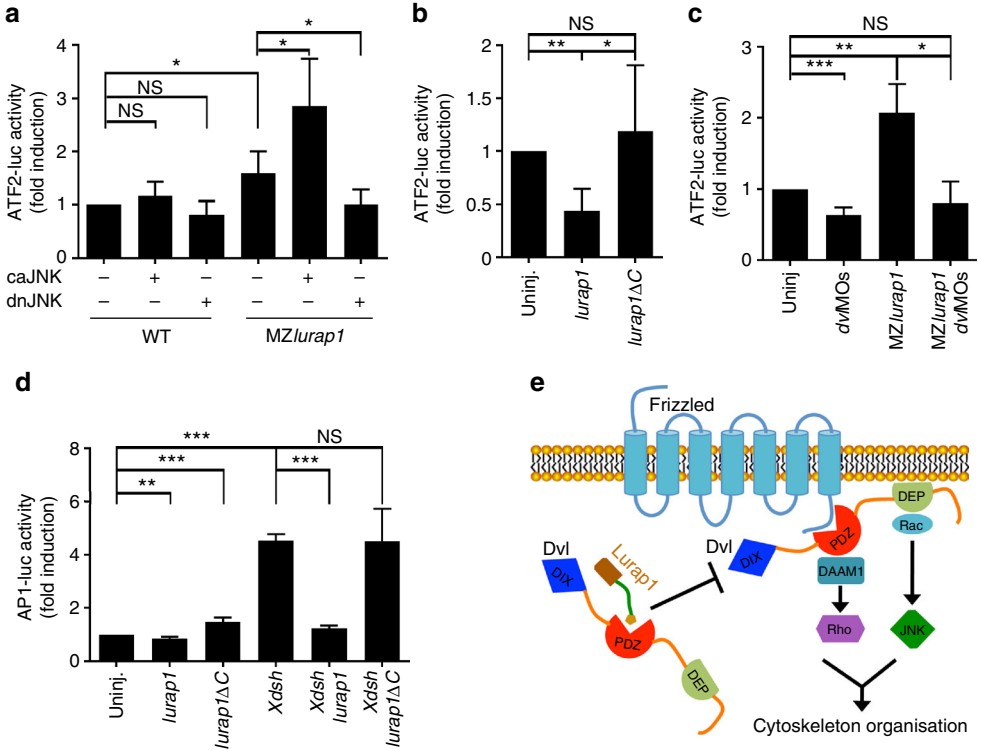

**Fig. 9** Lurap1 interacts with Dvl and JNK in Wnt/PCP signalling. The ATF2 and AP1 luciferase reporters were used to monitor Wnt/PCP activity changes in zebrafish and *Xenopus* embryos, respectively. **a** ATF2-luciferase reporter assay in WT and MZ*lurap1* embryos at 80% epiboly stage. An increase of ATF2 reporter activity is detected in MZ*lurap1* embryos, which is enhanced by caJNK, and blocked by dnJNK. Bars represent the mean values ± s.d. from three independent experiments (*P < 0.05; NS, not significant; Student's *t*-test). **b** Overexpression of Lurap1, but not Lurap1ΔC, inhibits the ATF2 reporter activity in WT zebrafish embryos. Bars represent the mean values ± s.d. from six independent experiments (*P < 0.05; **P < 0.01; NS, not significant; Student's *t*-test). **c** Knockdown of *dvl2* and *dvl3a* suppresses ATF2 reporter activation in WT and MZ*lurap1* embryos. Bars represent the mean values ± s.d. from three independent experiments (*P < 0.05; **P < 0.01; ***P < 0.01; NS, not significant; Student's *t*-test). **d** Analysis of the functional interaction between Lurap1 and Dvl using AP1 reporter in *Xenopus* dorsal mesoderm explants. Overexpression of Lurap1, but not Lurap1ΔC, suppresses Dvl-mediated increase of AP1 reporter activity. Bars represent the mean values ± s.d. from three independent experiments (**P < 0.01; ***P < 0.001; NS, not significant; Student's *t*-test). **e** A proposed model of Lurap1 function in regulating Wnt/PCP signalling and cell polarity. Localised cytoplasmic Lurap1 binds to the PDZ domain of Dvl through its PDZ-binding motif, and prevents Dvl from being recruited to the plasma membrane to activate downstream effectors

a low dose of *caJNK* mRNA or *dnJNK* mRNA in WT zebrafish embryos did not change ATF2 reporter activity. However, we observed an increase of ATF2 reporter activity in MZ*lurap1* embryos, which was enhanced by caJNK, but was suppressed by dnJNK (Fig. 9a). Conversely, overexpression of Lurap1, but not Lurap1ΔC, in WT embryos reduced ATF2 reporter activity by about twofold (Fig. 9b). These results indicate that loss-of-function of Lurap1 leads to an increased JNK activity. To determine whether Dvl mediates JNK activation in MZ*lurap1* embryos, we performed knockdown of Dvl2 and Dvl3a using low amounts of the corresponding morpholinos (1 and 2.5 ng per embryo, respectively). It showed that reducing Dvl dosage inhibited ATF2 reporter activity in WT embryos, and suppressed the increase of the reporter activity in MZ*lurap1* embryos (Fig. 9c). Thus, these observations suggest that the increased JNK activation following Lurap1 loss-of-function is at least partially mediated by Dvl.

We next analysed how Lurap1 regulates Dvl function in JNK activation in *Xenopus* embryos by using the AP1 luciferase reporter to monitor JNK activity[51]. Overexpression of Lurap1 in the dorsal mesoderm by injecting the corresponding mRNA (400 pg) decreased AP1 reporter activity, whereas injection of the same amounts of *lurap1ΔC* mRNA resulted in an increase of the reporter activity (Fig. 9d). This is in contrast with the absence of effect of Lurap1ΔC in zebrafish, and may be due to the reporter used and/or the targeted injection in the dorsal region in *Xenopus*

embryos. Nevertheless, we noticed that overexpression of Lurap1ΔC in zebrafish embryos also led to an increased ATF2 reporter activity in some experiments (see Fig. 9b). Single injection of *Xdsh* mRNA (200 pg) strongly increased AP1 reporter activity, which was suppressed by Lurap1, but not by Lurap1ΔC (Fig. 9d), indicating that Lurap1 specifically inhibits Dvl-induced JNK activation. Taken together, our results strongly suggest a functional interaction between Lurap1 and Dvl, which acts upstream of JNK to modulate Wnt/PCP signalling during CE movements (Fig. 9e). This function should also be conserved in vertebrates.

## Discussion

Wnt/PCP signalling plays a critical role in orchestrating asymmetric cell movements both in invertebrate and vertebrate embryos. Dvl is the key component of this pathway, which relays extracellular signal to downstream effectors, but how its activity is regulated remains unclear. Lurap1 has been identified as a molecular scaffold required for cell migration[40]. In this study, we have uncovered a role for this protein in Wnt/PCP signalling. We demonstrated that it is implicated in CE movements through interaction with Dvl, thereby influencing JNK activation. Our findings thus identify a Dvl-interacting partner, and provide further insight into the mechanism regulating Wnt/PCP pathway activation and polarised cell behaviours during morphogenetic movements.

Although Lurap1 plays an important role in cell migration, its function in morphogenetic movements during early development has not been studied. Our phenotype and time-lapse analyses indicate that MZ*lurap1* mutants bearing targeted mutation in the second exon display specific CE defects due to impaired cell polarity, suggesting that maternal Lurap1 is required for directional cell movements. We also generated mutations in a different region of the second exon as well as in the third exon, and observed the same phenotypes in MZ mutant embryos. In addition, introduction of the full-length Lurap1 in MZ*lurap1* embryos reduces the extent of CE defects. Thus, it can be concluded that the defective CE phenotypes are specifically caused by the inactivation of *lurap1* gene. Furthermore, overexpression of Lurap1 produces characteristic CE phenotypes in zebrafish and *Xenopus* embryos, suggesting a conserved function of Lurap1 in morphogenetic movements.

The CE phenotypes resulted from Lurap1 loss-of-function and gain-of-function are reminiscent of an altered activity of Wnt/PCP signalling, suggesting a possible implication of Lurap1 in this pathway. Indeed, we find a physical interaction between Lurap1 and Dvl, which involves the PDZ-binding motif of Lurap1 and the PDZ domain of Dvl. Moreover, Lurap1 and Dvl functionally coordinate polarised cellular activity. Live time-lapse imaging indicates that MZ*lurap1* mutant cells lose stable bipolar cellular protrusions, and become multipolar protrusive. Similarly, overexpression of Dvl or Xdd1 also leads to multipolar protrusive activity in zebrafish notochord cells, consistent with previous observations using *Xenopus* dorsal mesoderm explants[36]. In addition, we find that knockdown of Dvl results in an absence of cellular protrusion, indicating that inhibition of Wnt/PCP signalling blocks protrusive activity. Interestingly, Xdd1 lacking the PDZ domain potently enhances defective CE phenotypes in MZ*lurap1* embryos. Conversely, the disrupted cell polarity in MZ*lurap1* mutants is rescued by reducing Dvl dosage. These suggest that Lurap1 loss-of-function and Dvl overexpression similarly alter polarised cell behaviours, and that Xdd1 should activate Wnt/PCP signalling[44,45]. Furthermore, Dvl membrane recruitment, which is required for activation of the Wnt/PCP pathway[35,37], is enhanced in MZ*lurap1* embryos, whereas overexpression of Lurap1, but not Lurap1ΔC, prevents Frizzled-mediated Dvl membrane recruitment. The DEP domain plays a central role in the membrane translocation of Dvl[35], and the PDZ domain also interacts with Frizzled receptors[52], which may stabilise Dvl membrane localisation. We find that myc-tagged Lurap1 is predominantly localised in the cytoplasm, this likely impedes Frizzled-dependent Dvl membrane recruitment. In addition, myc-tagged Lurap1 may have some degrees of localised distribution in the cytoplasm of movement cells, which is consistent with the dynamic localisation observed in cultured migrating cells[40]. This may confer a localised activity to Lurap1 in organising cellular polarisation. Thus, it is tempting to speculate that the dynamic localisation of Lurap1 helps to modulate local recruitment of Dvl to the membrane, thereby leading to differential Wnt/PCP signalling activity within the cell, and resulting in cellular polarisation. Consistent with this hypothesis, our present results suggest that Lurap1 negatively regulate Wnt/PCP signalling at the level of Dvl, mostly through its PDZ-binding motif. In the absence of Lurap1 function, Dvl membrane recruitment is enhanced. Thus, this work identifies Lurap1 as a Dvl-interacting partner implicated in CE movements.

Lurap1 loss-of-function and Xdd1 overexpression disrupt cell polarity in a similar manner as the activation of JNK signalling, which can be rescued by dnJNK. Moreover, the increase of JNK activity in MZ*lurap1* embryos is suppressed by reducing Dvl dosage. Conversely, Lurap1 blocks JNK activation induced by Dvl. These are in accordance with a large body of evidence showing that Dvl and JNK function to mediate the activation of Wnt/PCP signalling, and that JNK is a known downstream effector of the Wnt/PCP pathway[25,27,28,42,43]. Therefore, our results from loss-of-function and gain-of-function analyses are consistent to suggest that Lurap1 functions to antagonise the activity of Dvl, thereby preventing or limiting JNK activation. Furthermore, it is well established that Wnt/PCP signalling controls polarised cell behaviours by inducing changes to the cytoskeleton. The positioning of the centriole/MTOC, a structure from which microtubules emerge, is regulated by Wnt/PCP signalling[48,49]. In zebrafish, the biased position of MTOCs has been shown to be dependent on Wnt/PCP components including Knypek and Dvl[48]. We find that MZ*lurap1* embryos present strongly randomised positioning of MTOCs in neuroectoderm, and particularly in notochord cells. Inappropriate activation of JNK signalling produces the same effect. These imply that Lurap1 regulates microtubule cytoskeleton organisation in a similar manner as Wnt/PCP signalling. However, we could not at present exclude the possibility that Lurap1 regulates cell polarity independently of Dvl since it also interacts with other proteins such as MRCK and MYO18A that play a role in cell adhesion and migration[40,41]. In addition, it is well established that Wnt/PCP signal is transduced via Dvl to the small GTPases Rac and Rho, which activate distinct downstream effectors to modulate the actin cytoskeleton[33]. Because Lurap1 activates the Rac effector, MRCK, to modulate filopodia formation in cultured cells[40], this raises the possibility that it could regulate cell polarity at the level of Rho and/or Rac. Further manipulating the activity of these small GTPases in MZ*lurap1* embryos could help to determine whether they interact in cytoskeleton organisation.

Wnt/PCP signalling needs to be tightly regulated in order to coordinate cellular polarisation in asymmetric cell movements. At present, several Dvl-interacting proteins that antagonise its activity have been identified as important modulators of Wnt/PCP signalling and CE movements both in *Xenopus* and zebrafish. First, Prickle is a context-dependent agonist or antagonist of Wnt/PCP signalling, essential for gastrulation cell movements[53–55]. It binds to Dvl and suppresses JNK activation induced by a high dose of Dvl[53]. Furthermore, Prickle is localised essentially in the cytoplasm and its overexpression prevents Frizzled-mediated Dvl membrane recruitment both in *Drosophila* and in zebrafish[55,56]. In *Drosophila*, the antagonistic action of Prickle on Dvl membrane localisation has been proposed to represent a feedback mechanism that helps to generate asymmetric distribution of Wnt/PCP components in pupal wing cells[56]. In this regard, Lurap1 may function in much the same manner as Prickle to modulate Dvl activity in Wnt/PCP signalling. Second, Dact (Dapper) is another Dvl-interacting protein that modulates both Wnt/ß-catenin and Wnt/PCP signalling[57–59]. Like Lurap1, Dapper binds to the PDZ domain of Dvl through its C-terminus PDZ-binding motif in *Xenopus*[57]. Loss of Dact1 leads to altered distribution of Dvl and increased JNK activity in mouse embryonic fibroblasts[60]. Taken together, our results suggest that Lurap1 represents a Dvl-interacting protein implicated in the regulation of Wnt/PCP signalling and cell polarity during CE movements.

In conclusion, the findings reported in this study strongly suggest that Lurap1 physically and functionally interacts with Dvl in Wnt/PCP signalling during CE movements. It acts to restrict JNK activation likely by modulating Dvl subcellular localisation and signalling activity. This is required for cytoskeletal reorganisation and the establishment of cell polarity in asymmetric movements. Because the absence of effect on Wnt/ß-catenin signalling, it raises the possibility that, to some extent, Lurap1 may regulate the specificity of Dvl in the Wnt signalling pathways. Altogether, our results shed light on the regulation of Dvl

activity in key morphogenetic events. They should help to elucidate the molecular mechanism underlying certain abnormal cell behaviours associated with inappropriate activation of the Wnt/PCP pathway.

## Methods

**Zebrafish and *Xenopus* embryos**. Zebrafish embryos were maintained at 28.5 °C, and staged according to standard criteria[6]. The *vangl2^m209/+* mutant line has been previously identified and characterised[22,61]. Zygotic homozygous offspring were obtained by intercrosses between the heterozygous carriers, and selected at 90% epiboly by morphological criteria followed by genotyping using allele-specific PCR (wtF, 5′-GTGTGTCTGCCTGTGTCTTACT-3′; mutF, 5′-GTGTGTCTGCCTGT GTCTTACT-3′; R, 5′-GATAAACTCCTCCCCCAGGT-3′)[62]. Synchronously fertilised embryos were injected at 1-cell stage in the yolk. *Xenopus* embryos were obtained from females injected with 500 IU of human chorionic gonadotropin (Sigma-Aldrich) and artificially fertilised using minced testis[63]. They were injected at 4-cell stage in the dorsal equatorial region. Dorsal mesoderm explants were dissected at the early gastrula stage, and cultured to the late gastrula stage.

The animal experiments were approved by the Experimental Animal Committee of Shandong University, and performed in accordance with the regulations and guidelines.

**Expression constructs and morpholinos**. The cDNA sequences coding for zebrafish Lurap1, Lurap1ΔC, Histone2B, and centrin4 were PCR-amplified, and cloned in pCS2 vector to generate N-terminally Flag-Lurap1, Flag-Lurap1ΔC, and myc-Lurap1, and C-terminally tagged Histone2B-RFP and centrin4-GFP. Other constructs including Xdsh, Xdd1, Xfz7, mGFP, mRFP, caJNK, and dnJNK were reported previously[18,20,44,64]. Capped mRNAs were synthesised from linearised plasmids by in vitro transcription using appropriate RNA polymerase.

The translation-blocking morpholinos against zebrafish *dvl2* (5′-GCGAAAGT ATGGAAGAGGTAGCGGC-3′), and *dvl3a* (5′-GGTAGATAACTTTAGTCTCCC CCAT-3′) were synthesised by Gene-Tools and dissolved in sterile water.

**TALEN constructs and targeted gene mutations in zebrafish**. TALEN repeat variable di-residues targeting sequences were assembled through Golden Gate Assembly[65], using the Golden Gate TALEN and TAL Effector Kit (#1000000016) from Addgene, and were cloned into modified pCS2+KKR and pCS2+ELD vectors[66]. The two TALEN mRNAs were mixed at equal amounts (200 pg each), and injected into 1-cell stage embryos. The targeting efficiency was determined by Sanger sequencing of genomic DNA extracted from 15 randomly selected embryos at 24 hpf. When the result indicates a positive targeting effect, injected embryos were reared to adulthood for outcrosses to screen F1 heterozygotes using genomic DNA extracted from tail fins.

**UV-mediated photo-conversion**. Synthetic mRNA (200 pg) encoding Kaede GFP was coinjected with *Xdd1* mRNA (80 pg) at 1-cell stage, and UV irradiation of the dorsal or lateral margin was performed at shield stage, with a DAPI filter using the smallest pinhole under a 20× objective of the microscope (Leica, LM2500)[67,68]. Images were taken at shield and bud stages to analyse the extent of CE.

**Cell transplantation**. Donor embryos were labelled with mGFP and Histone2B-RFP, or with centrin4-GFP, Histone2B-RFP, and mRFP, by injecting the corresponding mRNAs (300 pg) at 1-cell stage, and allowed to develop to sphere stage. Labelled blastoderm cells were aspirated with an injection micropipette and transplanted to the margin of an unlabelled recipient at the same stage. Chimeric embryos were cultured to appropriate stages for time-lapse analyses of cellular protrusions or MTOC positioning.

**Time-lapse imaging**. Embryos at 90% epiboly stage were mounted in a cavity microscope slide in 1% low-melting agarose. Cell movements were recorded using an upright light microscope (Leica, LM2500) or a High-Content Imaging System (MD, ImageXpress Micro-4), under differential interference contrast or epifluorescent illumination. The embryos were imaged every 15 or 30 s for a period of 15 or 30 min. Time-lapse movies were generated using ImageJ software (NIH Image).

**Immunofluorescence and confocal microscopy**. For Dvl membrane recruitment, each 1-cell stage zebrafish embryo was injected with *Xdsh* mRNA (100 pg) alone or coinjected with *Xfz7* mRNA (150 pg), with or without *lurap1* or *lurap1ΔC* mRNA (200 pg). The embryos were fixed at 40% epiboly stage and incubated in mouse monoclonal 9E10 anti-myc antibody (1:500, Santa Cruz Biotechnology, sc-40), followed by Alexa-488 conjugated secondary antibody (1:1000, INTERCHIM, FP-SA4010). The samples were analysed under a confocal microscope (Zeiss, LSM700).

Notochord cell shape and orientation were analysed at the tail bud stage by confocal microscopy following injection of synthetic mRNA encoding mGFP alone or together with *lurap1* or *lurap1ΔC* mRNA. The LWR of notochord cells and the

angles of the cell long axis relative to a line perpendicular to the anteroposterior axis of the notochord were calculated by ImageJ software.

To analyse the positioning of MTOCs, zebrafish embryos were labelled with centrin4-GFP and mRFP by injecting the corresponding mRNAs (300 pg) at 1-cell stage, and were fixed at 90% epiboly. Following DAPI staining, images were collected by confocal microscopy. The location of MTOCs relative to the nucleus in different germ layers was determined by measuring the angle formed between the line extending from the geometric centre of the nucleus to the animal pole direction and in parallel to the anteroposterior axis of the embryo, relative to the line connecting the centriole to the geometric centre of the nucleus, in a counter clockwise direction[48], and analysed using ImageJ software.

**Co-immunoprecipitation**. HEK293T cells (ATCC, CRL1573) were transfected with expression vectors using Lipofectamine 2000 (Invitrogen), and cultured for 24 h. The cells were washed with phosphate-buffered saline at room temperature, and lysed in ice-cold lysis buffer consisting of 150 mM NaCl, 50 mM Tris-HCl, pH 7.5, 1% (vol/vol) Triton X-100, 1 mM PMSF, and 1× protease inhibitor cocktail (Sigma-Aldrich). Immunoprecipitation was performed using 10 μl, monoclonal anti-myc agarose beads (Pierce, #20168) according to the manufacturer's recommendation. After 2 h of incubation at 4 °C, immunoprecipitated proteins were washed five times with washing buffer (a modified lysis buffer containing 500 mM NaCl), and separated by polyacrylamide gel electrophoresis, then transferred to nitrocellulose membrane, probed with 9E10 anti-myc (1:1000, Santa Cruz Biotechnology, sc-40) or anti-Flag (1:3000, Sigma-Aldrich, F3165) antibody, and detected using the Western-Lightning Plus-ECL substrate (PerkinElmer). All uncropped western blots can be found in Supplementary Fig. 6.

**In situ hybridisation and RT-PCR**. Whole-mount in situ hybridisation was performed according to standard protocol[69]. Zebrafish *lurap1* coding sequence was PCR amplified and cloned into pGEM-T Easy vector (Promega). This and other antisense probes, including *dlx3*, *ntla*, *papc*, *chordin*, *eve1*, *pax2a*, *krox20*, and *myod1*, were labelled using digoxigenin-11-UTP (Roche Diagnostics) and an appropriate RNA polymerase.

Total RNAs were extracted from zebrafish embryos at 24 hpf and reverse transcribed using M-MLV reverse transcriptase (Invitrogen). WT and mutant *lurap1* transcripts were PCR amplified using common primers (5′-TCATGGAGGAGAGTAATAATAC-3′ and 5′-ATCACAAAAACGTTACATCA TCCTG-3′), and genotyped by Sanger sequencing.

**Luciferase assays**. Zebrafish embryos at 1-cell stage were injected with 50 pg ATF2 reporter DNA, along with 5 pg pRL-TK DNA as an internal control. Twenty embryos at 80% epiboly were manually dechorionated and lysed in 60 μl lysis buffer (Promega). The lysate was clarified by centrifugation and luciferase activities were measured using a dual-luciferase reporter assay kit (Promega) according to the manufacturer's instructions. The AP1 reporter DNA (100 pg) was injected alone or coinjected with different synthetic mRNAs in the dorsal equatorial region of 4-cell stage *Xenopus* embryos, and measurement of AP1 luciferase activity using 10 dorsal mesoderm explants for each condition was done as above. The values were normalised with respect to Renilla luciferase activities, and the results were expressed as relative luciferase activities to control embryos.

**Statistical analyses**. All statistical data were obtained from at least three independent experiments using different batches of embryos, and analysed using the unpaired Student's t-test. P-values less than 0.05 and 0.01 were considered as significant (*) and extremely significant (**), respectively.

**Data availability**. The authors declare that all data supporting the findings of this study are available within the article and its Supplementary Information files or from the corresponding author upon reasonable request.

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

## Acknowledgements

We would like to thank S. Sokol, C. Niehrs, C.H.K. Cheng, B. Thisse, S.C. Lin, C. Vesque for providing the plasmid constructs and the zebrafish line used in this study, and members of the laboratory for technical assistance and zebrafish care. We also thank T.Y. Quan, H.Y. Yu, and X.M. Zhao for confocal and live image acquisition. This research was supported by Shandong University, the NSFC grants (31371355, 31471360, 31671509), and the Groupement des Entreprises Françaises dans la Lutte contre le Cancer (GEFLUC Paris).

## Author contributions

X.-N.C., M.S., J.-T.L., and J. Q. performed the zebrafish experiments and analysed the data; Y.-F.W. and Z.-G.X. realised biochemical experiments; D.-L.S. conceived the experiments, performed the *Xenopus* experiments, analysed the data and wrote the paper.

## Additional information

**Competing interests:** The authors declare no competing financial interests.

