## [Peer Review File · Nature Communications]

Reviewers' Comments:

Reviewer #1 (Remarks to the Author)

This paper by the group of Deli Shi addresses the role of Leucine Zipper adapter protein 1 (Lurap1) during convergent extension, using a combination of the zebrafish and *Xenopus* models. The authors propose that Lurap1 is involved in the regulation of Dishevelled (Dsh/Dvl) membrane localization/association and hence provides a new link to PCP signaling. Overall the data are presented adequately with some exceptions (see below) and the paper is worth considering for *Nature Communications*

There are however several problems with the paper as presented that need to be addressed before the paper can be considered publishable.

The main problem is in the interpretation of the interactions data between Dsh/Dvl and Lurap1. On page 8, bottom paragraph, the authors argue that the "interaction" between as seen in Xdd1 injected embryos that lack Lurap1 (MZLurap1 mutants) suggests that they act together. This is completely wrong however. As there is no Lurap1 present in such embryos, the observation that Xdd1 (a dominant negative Dsh/Dvl isoform that – based on the authors data - does not bind Lurap1 anymore) enhances the Lurap1 null mutant embryos can only be interpreted that the Xdd1 effect – and hence action of Dsh/Dvl is Lurap1 independent. So the two proteins have to act in PARALLEL, period. This needs to be addressed and discussed properly and the whole interpretation of the function of Lurap1 has thus to be changed.

Related to this, the authors should address how Lurap1 is localized in wild-type, Dsh/Dvl mutants, and other PCP signaling mutants (e.g. the Vangl2 trilobite MZ-null springs to mind as a good genotype to test this in, as it shows no redundancy with other PCP factors).

Other comments:

1. While the data presentation overall is adequate, there are some instances where more data needs to be added. For example Figure 6 shows one panel with an MTOC localization and the rest is all quantifications. Show more original data. More and better panels of MTOC localization samples need to be presented. Similarly, statistical analyses are not always present. For example Figure 5o does not have any statistics.
2. There are many, many typos throughout the text. Also the paper should be carefully edited by a native English speaker, it is clumsy in several places.
3. On page 6 the authors refer to the phenotype as a "delay". Is it really just a delay? To me it looks like a strong defective phenotype of CE aspects of gastrulation, not just a delay. Please clarify.
4. Line 58: Boutros et al 1998 reference is missing here – it was the first paper to make a link between PCP-Dsh function and JNK activation.
5. There are better and more recent reviews for the Frizzled-Dsh/Dvl-PCP pathway. The authors should consider replacing/updating those review references.

Reviewer #2 (Remarks to the Author)

The manuscript by Li et al. describes the role of leucine repeat adaptor protein1 (Lurap1), a novel

Dishevelled-binding protein that acts in the non-canonical Wnt/planar cell polarity (PCP) pathway, in the regulation of convergence and extension (CE) movements during vertebrate gastrulation. The authors show that maternal-zygotic null mutants for zebrafish *lurap1* (MZ*lurap1*) exhibit defective CE movements, primarily due to activation of Dishevelled and JNK. Also, the authors show that the positioning of microtubule-organizing center (MTOC) of MZ*lurap1* cells is disrupted.

Although the authors found an upstream negative regulator of Dishevelled, there is no novel mechanistic insight into what aspects of CE cell behaviors are mediated by Dishevelled and JNK, respectively. In particular, JNK-dependent CE cell behavior is unclear from this study as well as in the previous studies, as the analysis has not been done based on live imaging. The CE analysis in this study is done superficially based on fixed embryos/tissues, so that the differences in “no polarity” and “hyper-polarity” cannot be resolved as both would lead to similar CE defects apparently. Therefore, this work lacks general attraction of broad readership in cell/developmental biologists.

(Major points)

In the over-simplified view, the authors emphasize that over-activation of Dishevelled (its membrane localization) causes activation of JNK in MZ*lurap1* embryos. But, activation of Dishevelled or JNK might be regulated differently by *lurap1*. The authors need to demonstrate this point clearly based on *in vivo* time-lapse analysis.

The major problem is the interpretation of the effect of the *Xdd1* on MZ*lurap1* embryos in Fig3. Previously, it has been used to block the Wnt/PCP pathway (by suppressing polarized cell behavior). But, now the authors use/interpret this to activate the JNK pathway. Instead of using *Xdd1*, the MZ*lurap1* phenotype (over-activation of Dishevelled) should be rescued by reducing Dishevelled dosage.

In Fig7, it is unclear what aspect of CE cell behavior of MZ*lurap1* is rescued by dnJNK.

The analysis of MTOC underlying CE ought be done in cells of the ectoderm/mesoderm at 90% epiboly based on live-imaging analysis (Sepich, 2011), rather than EVL/deep cells at shield stage (as those cells are not undergoing CE). Do the embryos with increased JNK activity (e.g. expressing CA-JNK) show the MTOC phenotype, similar to MZ*lurap1*?

Responses to reviewers:

We have thoroughly considered the reviewer's comments and made substantial changes in the revised manuscript. Due to the extensive revision and the amount of work done to perform additional experiments, the order of contribution authors is changed. We thank the reviewers for their helpful comments, and hope that the issues are addressed in a satisfactory manner.

Reviewer #1 (Remarks to the Author):

This paper by the group of Deli Shi addresses the role of Leucine Zipper adapter protein 1 (Lurap1) during convergent extension, using a combination of the zebrafish and *Xenopus* models. The authors propose that Lurap1 is involved in the regulation of Dishevelled (Dsh/Dvl) membrane localization/association and hence provides a new link to PCP signaling. Overall the data are presented adequately with some exceptions (see below) and the paper is worth considering for Nature Communications

There are however several problems with the paper as presented that need to be addressed before the paper can be considered publishable.

The main problem is in the interpretation of the interactions data between Dsh/Dvl and Lurap1. On page 8, bottom paragraph, the authors argue that the "interaction" between as seen in Xdd1 injected embryos that lack Lurap1 (MZLurap1 mutants) suggests that they act together. This is completely wrong however. As there is no Lurap1 present in such embryos, the observation that Xdd1 (a dominant negative Dsh/Dvl isoform that – based on the authors data - does not bind Lurap1 anymore) enhances the Lurap1 null mutant embryos can only be interpreted that the Xdd1 effect – and hence action of Dsh/Dvl is Lurap1 independent. So the two proteins have to act in PARALLEL, period. This needs to be addressed and discussed properly and the whole interpretation of the function of Lurap1 has thus to be changed.

We thank this reviewer for insightful comments that help us to improve the manuscript. Indeed, in the previous version, there was no sufficient evidence showing that Lurap1 and Dvl function together in Wnt/PCP signalling. Since Lurap1 binds to Dvl, and they functionally interacts in regulating Wnt/PCP signalling and cell polarity, we suggest that Lurap1 antagonises Dvl activity, and that the absence of Lurap1 function leads to increased Dvl activity in Wnt/PCP signalling. This perturbs polarised cell behaviours during CE movements. In this revised version, we present several experimental results supporting this conclusion. First, both Lurap1 loss-of-function and Dvl or Xdd1 overexpression impair cell polarity in the same manner, and the effect of Lurap1 loss-of-function can be rescued by reducing Dvl dosage (Figure 4). Second, in MZLurap1 mutant

cells, *Dvl* membrane localisation is enhanced. Conversely, overexpression of *Lurap1*, but not *Lurap1 Δ C*, prevents *Dvl* membrane recruitment (Figure 5). Since *Dvl* membrane localisation has been shown to be required for *Wnt/PCP* signalling (Park et al., 2005), this observation suggests that *Lurap1* should antagonise *Dvl* activity in *Wnt/PCP* signalling. Finally, *Lurap1* also inhibits *Dvl*-induced *JNK* activation, and there is an increased *JNK* activity in *MZlurap1* mutants, which can be rescued by reducing *Dvl* dosage (Figure 8). Based on these data, we conclude that *Lurap1* plays a role in modulating *Dvl* activity in *Wnt/PCP* signalling. This is discussed in the revised manuscript. We also do not exclude the possibility that *Lurap1* and *Dvl* function in parallel or independently, this point is addressed in the Discussion (Page 16).

Related to this, the authors should address how *Lurap1* is localized in wild-type, *Dsh/Dvl* mutants, and other *PCP* signaling mutants (e.g. the *Vangl2* trilobite *MZ*-null springs to mind as a good genotype to test this in, as it shows no redundancy with other *PCP* factors).

We followed this suggestion and think that it is an important issue. So we examined *Lurap1* subcellular localisation in different germ layers (neuroectoderm, paraxial mesoderm, and notochord) engaged in *CE* movements. This experiment was performed both in wild-type and *Wnt/PCP* signalling defective embryos using N-terminally myc-tagged *Lurap1* that was found to be more appropriate for this analysis. At present, no *Dvl* mutants are available in zebrafish and probably it may be necessary to generate double homozygous mutants due to the presence of multiple *Dvl* isoforms. We used *Dvl2* and *Dvl3a* morpholinos to knockdown these genes that are most abundantly expressed at early stages (Harvey et al., 2013). Indeed, we obtained specific *C&E* phenotypes (Supplementary Figure 7). The results of *Lurap1* subcellular localisation are presented in Supplementary Figure 8, they show that *Lurap1* display some degree of localised distribution in *WT* embryos, and this becomes less evident in *Dvl* morphants and in homozygous *vang2/trilobite* mutants.

Other comments:

1. While the data presentation overall is adequate, there are some instances where more data needs to be added. For example Figure 6 shows one panel with an MTOC localization and the rest is all quantifications. Show more original data. More and better panels of MTOC localization samples need to be presented. Similarly, statistical analyses are not always present. For example Figure 5o does not have any statistics.

We fully agree with the comment on MTOC localisation data, and repeated the experiments according to previously reported methods (Sepich et al., 2011). In the revised version, MTOC

positioning was examined both in neuroectoderm and notochord of WT and MZlurap1 embryos, as well as embryos overexpressing caJNK. The original data, along with their quantifications, are presented in Figure 7 and Supplementary movies 17-19.

We also made the necessary efforts to present statistical analyses appropriately using a large number of samples. These are either indicated in the Figures or described in the text. For Figure 5o in the previous version (Figure 6o in this revised version), we would like to clarify that it was used to statistically present different categories of CE defective phenotypes. We modified the presentation of this Figure, and described more explicitly in the legend that the colour codes for the stacked column are indicated in the live images. We hope that this graph is appropriate for this kind of analysis.

2. There are many, many typos throughout the text. Also the paper should be carefully edited by a native English speaker, it is clumsy in several places.

We have made all the efforts to minimise typos and to edit the text, and asked our colleagues for critical reading. We hope that these helped to improve the revised manuscript.

3. On page 6 the authors refer to the phenotype as a “delay”. Is it really just a delay? To me it looks like a strong defective phenotype of CE aspects of gastrulation, not just a delay. Please clarify.

Thanks for this comment. Indeed, referring the C&E phenotype as a “delay” is not appropriate. We therefore replaced it by “defects”. In addition, to demonstrate this more rigorously we examined the width of paraxial mesoderm after in situ hybridisation in flat-mounted embryos. The results show that the paraxial mesoderm is significantly wider in MZlurap1 mutants compared to WT embryos with an equal number of somites, indicating that the CE defects are not due to a developmental delay (Supplementary Fig. 4).

4. Line 58: Boutros et al 1998 reference is missing here – it was the first paper to make a link between PCP-Dsh function and JNK activation.

This is included in the revised manuscript.

5. There are better and more recent reviews for the Frizzled-Dsh/Dvl-PCP pathway. The authors should consider replacing/updating those review references.

We updated the review references on Frizzled/Dvl function in Wnt/PCP pathway. These include Alder et al., 2012; Singh & Mlodzik, 2012; Yang & Mlodzik, 2015; Sokol, 2015; Wang et al., 2016; Jussil & Ciruna, 2017; Butler & Wallingford, 2017.

Reviewer #2 (Remarks to the Author):

The manuscript by Li et al. describes the role of leucine repeat adaptor protein1 (Lurap1), a novel Dishevelled-binding protein that acts in the non-canonical Wnt/planar cell polarity (PCP) pathway, in the regulation of convergence and extension (CE) movements during vertebrate gastrulation. The authors show that maternal-zygotic null mutants for zebrafish lurap1 (MZlurap1) exhibit defective CE movements, primarily due to activation of Dishevelled and JNK. Also, the authors show that the positioning of microtubule-organizing center (MTOC) of MZlurap1 cells is disrupted.

Although the authors found an upstream negative regulator of Dishevelled, there is no novel mechanistic insight into what aspects of CE cell behaviors are mediated by Dishevelled and JNK, respectively. In particular, JNK-dependent CE cell behavior is unclear from this study as well as in the previous studies, as the analysis has not been done based on live imaging. The CE analysis in this study is done superficially based on fixed embryos/tissues, so that the differences in “no polarity” and “hyper-polarity” cannot be resolved as both would lead to similar CE defects apparently. Therefore, this work lacks general attraction of broad readership in cell/developmental biologists.

We thank the reviewer for this important criticism, and agree that both Dvl- and JNK-dependent CE cell behaviours need an in-depth analysis. We have made all the efforts to address these issues by using time-lapse imaging. A large number of new experiments have been performed during the revision. The analyses include CE movements in whole embryos, polarised protrusive activity in transplanted cells, and the positioning of MTOCs in different germ layers undergoing CE movements. Additional data are presented in Figures 2, 4 and 7, and Supplementary Figure 5. We also include 19 Supplementary movies associated with these Figures, which should be useful for the scientific community. To summarize, we show that MZlurap1 mutants display impaired CE cell behaviour with reduced extent of convergence and extension because the cells move along zigzagging trajectories (Figure 2). We also show that MZlurap1 mutant cells, Dvl-, Xdd1- or caJNK-overexpressing cells are multipolar protrusive, whereas knockdown or blockade of JNK signalling leads to absence of cellular protrusions. Furthermore, the abnormal cell polarity in MZlurap1 mutant cells is rescued by reducing Dvl dosage or by dnJNK (Figure 4). Finally, we show that Lurap1 loss-of-function and overexpression of caJNK similarly disrupt the biased positioning of MTOCs in neuroectoderm and notochord cells at 90% epiboly (Figure 7). We hope that these novel results could improve the analyses of Lurap1, Dvl and JNK function in CE cell behaviours, and provide mechanistic insight into the regulation of cell polarity during CE movements.

(Major points)

In the over-simplified view, the authors emphasize that over-activation of Dishevelled (its membrane localization) causes activation of JNK in MZlurap1 embryos. But, activation of Dishevelled or JNK might be regulated differently by lurap1. The authors need to demonstrate this point clearly based on in vivo time-lapse analysis.

We totally agree with this comment and toned down the claim in the revised manuscript. Indeed, we do not have direct evidence demonstrating that an enhanced Dvl membrane localisation causes activation of JNK signalling in MZlurap1 embryos, although it was previously reported that Dvl membrane localisation is required for Wnt/PCP signalling in Xenopus (Park et al., 2005). Thus, we clarified this point by presenting the enhanced Dvl membrane localisation and increased JNK activity as associated with the disrupted cell polarity in MZlurap1 embryos. The Abstract is modified accordingly.

We also performed in vivo time-lapse imaging to analyse the functional interaction between Lurap1, Dvl and JNK signalling. The results indicate that dnJNK is able to suppress the multipolar protrusive activity in MZlurap1 and Xdd1-overexpressing embryos (Figure 4). To some extent, this suggests that Lurap1 and Dvl regulate cell polarity upstream of JNK signalling. It also indicates that overexpression of Xdd1 should activate JNK signalling.

The major problem is the interpretation of the effect of the Xdd1 on MZlurap1 embryos in Fig3. Previously, it has been used to block the Wnt/PCP pathway (by suppressing polarized cell behavior). But, now the authors use/interpret this to activate the JNK pathway. Instead of using Xdd1, the MZlurap1 phenotype (over-activation of Dishevelled) should be rescued by reducing Dishevelled dosage.

Thanks for this helpful suggestion. We performed the experiment using time-lapse imaging. As discussed above, both Lurap1 loss-of-function and Xdd1 overexpression suppressed polarised cell behaviour by inducing multipolar protrusive activity. This is consistent with previous report in Xenopus embryos (Wallingford et al., 2000). We further show that the impaired cell polarity in MZlurap1 embryos is rescued by partial knockdown of Dvl2 and Dvl3a. These new data are presented in Figure 4, and the previous Figure 3 (Xdd1 in MZlurap1 embryos) is presented as Supplementary Figure 6 to emphasize the functional interaction between Lurap1 and Dvl by an alternative approach. In addition, by reporter assay we show that the increased JNK activity in MZlurap1 embryos is suppressed by reducing Dvl dosage (Figure 8). These suggest an increased activity of Dvl in Wnt/PCP signalling following Lurap1 loss-of-function.

In Fig7, it is unclear what aspect of CE cell behavior of MZlurap1 is rescued by dnJNK.

Since we show in Figure 4 that the polarised protrusive activity of MZlurap1 mutant cells is rescued by dnJNK, the previous Figure 7a-d is presented as Supplementary Figure 9 to show statistically the functional interaction between Lurap1 and JNK.

The analysis of MTOC underlying CE ought be done in cells of the ectoderm/mesoderm at 90% epiboly based on live-imaging analysis (Sepich, 2011), rather than EVL/deep cells at shield stage (as those cells are not undergoing CE). Do the embryos with increased JNK activity (e.g. expressing CA-JNK) show the MTOC phenotype, similar to MZlurap1?

We thoroughly followed these suggestions and performed the experiments according to previously reported methods (Sepich et al., 2011). The positioning of MTOCs in neuroectoderm and notochord cells was analysed both by time-lapse imaging and in fixed embryos at 90% epiboly. The results show that the biased location of MTOCs is disrupted in MZlurap1 mutants, as well as in embryos overexpressing caJNK, in much the same way. These new data are presented in Figure 7 and Supplementary movies 17-19. The analysis of MTOCs at shield stage is removed from the revised manuscript.

Reviewers' Comments:

Reviewer #1:

Remarks to the Author:

The revised paper by the lab of De-li Shi is significantly improved. In my view the authors have addressed my comments adequately. I also think they improved the written language and have addressed the comments from the other reviewer. I recommend now publication of this paper.

Reviewer #2:

Remarks to the Author:

The revised manuscript by Cheng et al. has significantly improved after taking reviewers' comments into account.

However, the new data in Figure 4f-o and Movies 7-16 that present the regulation of cell polarity by Lurap1/Dsh/JNK are premature and rather confusing in relation to the summary diagram in Figure 8e. The limitation is that the pictures and movies do not always reflect the description in the main text and figure legends. The authors should quantitate the imaging data based on several different criteria (orientation/roundness of the cell, the type and orientation of the processes) then interpret what are rescued clearly. The quality of some of the pictures is too poor to distinguish between normal polarity and randomized polarity or between lamellipodia and filopodia. The authors need to clarify the following points:

a) In Figure 4g and movie 8 (MZlurap1), there are apparently two mixed cell populations: one that is able to stretch and orient properly with randomized processes (filopodia in the majority) and the other that is round with lots of blebs (increased motility), whilst stating in the text that "MZlurap1 mutant cells were not elongated. They were multipolar protrusive by forming multiple lamellipodia and filopodia in all directions".

b) The authors over-simplify in the text that "Overexpression of Xdsh or Xdd1 results in rounded multipolar protrusive cells." To this reviewer, the movies 9 (Xdsh) and 10 (Xdd1) do not look the same; Xdsh cells show predominant lamellipodia in random directions, whereas Xdd1 cells exhibit filopodia in random directions. In other words, Xdd1 activates only JNK (to produce filopodia), while Xdsh activates RhoA/Rac and JNK. The authors ought demonstrate whether Xdsh cells can be rescued by dn-JNK.

c) The model in Figure 8e is not supported by the data. The authors showed that the Lurap1-Xdd1-JNK axis mediates cell polarity but have not demonstrated the Lurap1-Dsh-RhoA axis. Since MZlurap1 cells are rescued just by dn-JNK, together with the fact that dvl2/dvl3 morphant cells show no protrusive activity, the simple interpretation is that Lurap1 could normally activate RhoA and/or Rac in a Dsh-independent manner. If the authors agree with this point, this possibility should be included in Discussion.

Response to reviewers:

Reviewers' comments:

Reviewer #1 (Remarks to the Author):

The revised paper by the lab of De-li Shi is significantly improved. In my view the authors have addressed my comments adequately. I also think they improved the written language and have addressed the comments from the other reviewer. I recommend now publication of this paper.

We are very pleased by the positive recommendation from this reviewer, and thank the reviewer for the expert review that helped to improve the manuscript.

Reviewer #2 (Remarks to the Author):

The revised manuscript by Cheng et al. has significantly improved after taking reviewers' comments into account.

However, the new data in Figure 4f-o and Movies 7-16 that present the regulation of cell polarity by Lurap1/Dsh/JNK are premature and rather confusing in relation to the summary diagram in Figure 8e. The limitation is that the pictures and movies do not always reflect the description in the main text and figure legends. The authors should quantitate the imaging data based on several different criteria (orientation/roundness of the cell, the type and orientation of the processes) then interpret what are rescued clearly. The quality of some of the pictures is too poor to distinguish between normal polarity and randomized polarity or between lamellipodia and filopodia. The authors need to clarify the following points;

We thank this reviewer for rigorous review and constructive comments on the presentation of the data. During the revision, we have endeavoured to make high quality images and movies. We also followed the reviewers' suggestions to statistically analyse changes in cell polarity using different criteria. As a result, Figure 4f-o and the corresponding movies are replaced by improved data, and the results from statistical analyses are presented in the new Figure 5, and the new Supplementary Figure 8. These new images and movies, along with the statistical analyses, clarified different cell behaviour changes, and further support our conclusion on the regulation of cell polarity by Lurap1/Dsh/JNK pathway.

a) In Figure 4g and movie 8 (MZlurap1), there are apparently two mixed cell populations: one that is able to stretch and orient properly with randomized processes (filopodia in the majority) and the other that is round with lots of blebs (increased motility), whilst stating in the text that "MZlurap1 mutant cells were not elongated. They were multipolar protrusive by forming multiple lamellipodia and filopodia in all directions".

This is a very interesting remark. Indeed, the cell transplantation approach that we used to better visualize cell polarity changes by live time-lapse imaging can result in two populations of descendent cells, according to their location in the notochord or in the ectoderm. We agree that those *MZlurap1* mutant cells that are integrated in the notochord are able to stretch and orient properly to some extent. However, they form randomized filopodia. Other cells located in the ectoderm are round with blebs in all directions. In the revised manuscript, we clarified these points by the following description that is highlighted in Page 8: “*MZlurap1* mutant cells in the notochord were not correctly aligned mediolaterally, and were less elongated. They displayed unstable multipolar protrusive activity by forming and retracting lamellipodia and especially very long filopodia in all directions (Fig. 4g; Supplementary Movie 8). Interestingly, *MZlurap1* cells located in the ectoderm were round with multiple blebs in randomised directions (Supplementary Movie 9), indicating increased, but undirected motility”. Thus, an additional movie is provided with the revised manuscript to illustrate the formation of blebs in *MZlurap1* mutant cells in the ectoderm.

b) The authors over-simplify in the text that “Overexpression of *Xdsh* or *Xdd1* results in rounded multipolar protrusive cells.” To this reviewer, the movies 9 (*Xdsh*) and 10 (*Xdd1*) do not look the same; *Xdsh* cells show predominant lamellipodia in random directions, whereas *Xdd1* cells exhibit filopodia in random directions. In other words, *Xdd1* activates only JNK (to produce filopodia), while *Xdsh* activates RhoA/Rac and JNK. The authors ought demonstrate whether *Xdsh* cells can be rescued by dn-JNK.

We agree that *Xdsh*- and *Xdd1*-induced cell polarity changes were not described sufficiently. The text is now modified in Page 8 as follows: “overexpression of *Xdsh* and *Xdd1* by injecting the corresponding mRNA (80 pg) in WT embryos produced multipolar protrusive cells. Compared to *Xdd1*, *Xdsh* induced more lamellipodia in all directions, which also extends filopodia, and *Xdd1* induced essentially randomised filopodia (Fig. 4h,i; Supplementary Movies 10, 11)”. The description of cell processes in different cells was also detailed in the Figure legend.

Moreover, we followed the reviewers’ suggestion and examined the rescue of cell polarity in *Xdsh*-expressing cells by dnJNK. The result is described in the text (Page 9). It indicates that randomized filopodia are largely rescued, and randomized lamellipodia are also rescued to a significant extent (Fig. 4p; Fig. 5; Supplementary Fig. 8).

c) The model in Figure 8e is not supported by the data. The authors showed that the *Lurap1*-*Xdd1*-JNK axis mediates cell polarity but have not demonstrated the *Lurap1*-Dsh-RhoA axis. Since *MZlurap1* cells are rescued just by dn-JNK, together with the fact that *dvl2/dvl3* morphant cells show no protrusive activity, the simple interpretation is that *Lurap1* could normally activate RhoA and/or Rac in a Dsh-independent manner. If the authors agree with this point, this

possibility should be included in Discussion.

Thanks for this remark and suggestion. It is well established that Wnt/PCP signalling is transduced via Dvl to Rho and Rac (Wallingford and Habas, 2005; reference cited in the manuscript). Although the Dsh/RhoA pathway was not addressed in this work, it was included in the model to fully represent the current knowledge on the cascades downstream of Dvl in Wnt/PCP signalling. For the Dsh/Rac pathway, our data summarized in the model suggest that Lurap1 regulates Dvl membrane localization (Fig. 5), and that it functionally interacts with Dvl in cell polarity and JNK signalling (Fig. 4, Fig. 9c,d). We would like to emphasize that the perturbed cell polarity in MZ*lurap1* cells is rescued by reducing Dvl dosage (Figure 4m, Supplementary Movie 15), and by dnJNK (Fig. 4n, Supplementary Movie 16). These suggest that Lurap1 functions at the level of Dvl, as proposed in the model. However, we could not exclude the possibility that Lurap1 might regulate RhoA and/or Rac independently of Dvl, given that it can activate the Rac1 effector, MRCK, to modulate filopodia formation in cultured cells (Tan et al., 2008; reference cited and discussed in the manuscript).

We thus followed the reviewers' suggestion, and included this point in the Discussion. A possibility that Lurap1 regulates cell polarity through interaction with other partners was mentioned briefly in the previous version (Page 16). We further discussed this point by including the following sentences: "In addition, it is well established that Wnt/PCP signal is transduced via Dvl to the small GTPases Rac and Rho, which activate distinct downstream effectors to modulate the actin cytoskeleton³³. Because Lurap1 activates the Rac effector, MRCK, to modulate filopodia formation in cultured cells⁴⁰, this raises the possibility that it could regulate cell polarity at the level of Rho and/or Rac. Further manipulating the activity of these small GTPases in MZ*lurap1* embryos could help to determine whether they interact in cytoskeleton organisation".

We feel that we have made the efforts to fully address the points raised by reviewer #2, and provided high quality images and movies in this revised version. We would to thank this reviewer for insightful comments that help us to improve the presentation and the quality of this manuscript.

Reviewers' Comments:

Reviewer #2:

Remarks to the Author:

The revised manuscript by Cheng et al. has been improved to a satisfactory degree. Now I recommend this for publication with a minor revision.

(Minor point)

In new Supplementary Figure 8 "Statistical analyses of the type and orientation of cellular protrusions", the columns of Filopodia and Lamellipodia need to be separated, and error bars are to be added on each. If there are significant differences between the groups, these should be indicated.

Response to reviewers:

REVIEWERS' COMMENTS:

Reviewer #2 (Remarks to the Author):

The revised manuscript by Cheng et al. has been improved to a satisfactory degree. Now I recommend this for publication with a minor revision.

(Minor point)

In new Supplementary Figure 8 “Statistical analyses of the type and orientation of cellular protrusions”, the columns of Filopodia and Lamellipodia need to be separated, and error bars are to be added on each. If there are significant differences between the groups, these should be indicated.

We thank this reviewer for recommendation of publication with minor revision, and the expert review to improve the manuscript. Supplementary Figure 8 is now split into two parts, and statistical analyses of filopodia and lamellipodia are separated. Error bars are added to each column, and significance between the groups is indicated.